EMBO
Molecular Medicine

# A novel epigenetic *AML1-ETO/THAP10/miR-383* mini-circuitry contributes to t(8;21) leukaemogenesis

Yonghui Li[1],[†] (ID), Qiaoyang Ning[1,2,†], Jinlong Shi[3,†], Yang Chen[4,†], Mengmeng Jiang[1], Li Gao[5], Wenrong Huang[1], Yu Jing[1], Sai Huang[1], Anqi Liu[1], Zhirui Hu[4], Daihong Liu[1], Lili Wang[1], Clara Nervi[6], Yun Dai[7,8], Michael Q Zhang[4] & Li Yu[1],* (ID)

## Abstract

DNA methylation patterns are frequently deregulated in t(8;21) acute myeloid leukaemia (AML), but little is known of the mechanisms by which specific gene sets become aberrantly methylated. Here, we found that the promoter DNA methylation signature of t(8;21)[+] AML blasts differs from that of t(8;21)[−] AMLs. This study demonstrated that a novel hypermethylated zinc finger-containing protein, THAP10, is a target gene and can be epigenetically suppressed by AML1-ETO at the transcriptional level in t(8;21) AML. Our findings also show that *THAP10* is a *bona fide* target of *miR-383* that can be epigenetically activated by the AML1-ETO recruiting co-activator p300. In this study, we demonstrated that epigenetic suppression of *THAP10* is the mechanistic link between AML1-ETO fusion proteins and tyrosine kinase cascades. In addition, we showed that THAP10 is a nuclear protein that inhibits myeloid proliferation and promotes differentiation both *in vitro* and *in vivo*. Altogether, our results revealed an unexpected and important epigenetic mini-circuit of *AML1-ETO/THAP10/miR-383* in t(8;21) AML, in which epigenetic suppression of *THAP10* predicts a poor clinical outcome and represents a novel therapeutic target.

**Keywords** *AML1-ETO*; epigenetics; *miR-383*; t(8;21) AML; *THAP10*
**Subject Categories** Cancer; Chromatin, Epigenetics, Genomics & Functional Genomics; Haematology

## Introduction

t(8;21) acute myeloid leukaemia (AML) is a highly heterogeneous disease from a biological and clinical standpoint (Hsiao *et al*, 2015). This remains a significant barrier to the development of accurate clinical classification, risk stratification and targeted therapy for this disease. Epigenetic control of gene expression has been suggested to play a pivotal role in the biological behaviour of cells (Wouters & Delwel, 2016). Disruption of normal DNA methylation distribution is a hallmark of cancer and has critical roles in the initiation, progression and maintenance of malignant phenotypes (Figueroa *et al*, 2010). Several tumour suppressor genes [e.g. *miR-193a* (Li *et al*, 2013b) and *miR-223* (Fazi *et al*, 2007a; Pulikkan *et al*, 2010)] have been shown to be abnormally methylated in t(8;21) AML patients. Moreover, aberrant distribution of promoter DNA methylation occurring in specific and distinct patterns was identified as a universal feature in t(8;21) AML (Figueroa *et al*, 2010). However, the mechanisms underlying these aberrant methyl cytosine patterns have not been defined.

The thanatos-associated proteins (THAPs), a novel family of cellular factors, are characterized by the presence of an evolutionarily conserved protein motif (Cayrol *et al*, 2007). Overall, there are 12 distinct human proteins that contain the THAP domain (THAP 0–11; Zhu *et al*, 2009). Among them, THAP1 has been identified as a nuclear protein associated with promyelocytic leukaemia nuclear bodies (PML NBs; Cayrol *et al*, 2007). THAP7 has both histone-binding and putative DNA-binding motifs (Macfarlan *et al*, 2005). THAP9 was shown to be an active DNA transposase and still retains the catalytic activity to mobilize *P* transposable elements across species (Majumdar *et al*, 2013). Most recently, the mouse homolog of THAP11, Ronin, was identified as an essential factor underlying embryogenesis and embryonic stem

1 Department of Haematology, Chinese PLA General Hospital, Beijing, China
2 Nankai University School of Medicine, Tianjin, China
3 Department of Biomedical Engineering, Chinese PLA General Hospital, Beijing, China
4 Key Laboratory of Bioinformatics, Tsinghua University, Beijing, China
5 Department of Haematology, China-Japan Friendship Hospital, Beijing, China
6 Department of Medico-Surgical Sciences and Biotechnologies, University of Rome "La Sapienza" Polo Pontino, Latina, Italy
7 Cancer Centre, The First Hospital of Jilin University, Changchun, China
8 Department of Internal Medicine, Massey Cancer Center, Virginia Commonwealth University, Richmond, VA, USA
*Corresponding author. Tel: +86 10 55499003; Fax: +86 10 68150721, E-mail: liyu301@vip.163.com
†These authors contributed equally to this work

cell pluripotency (Dejosez *et al*, 2008). In contrast, Zhu *et al* (2009) reported that THAP11 functions as a negative regulator of cell growth in human hepatoma cells through transcriptional repression of the proto-oncogene *MYC*. These studies, together with data obtained in *C. elegans*, indicated that THAPs are zinc-dependent, sequence-specific DNA-binding factors involved in cell proliferation, apoptosis, the cell cycle, chromatin modification and transcriptional regulation. However, the transcriptional regulatory properties of most human THAPs and their roles in pathological processes remain largely unknown.

Given the function of AML1-ETO as a transcriptional repressor (DeKelver *et al*, 2013), we investigated whether AML1-ETO was associated with aberrant epigenetic programming in t(8;21) AML. Here, we report an unexpected and functional epigenetic mini-circuit of *AML1-ETO/THAP10/miR-383* in t(8;21) AML, which might represent a novel therapeutic target as well as a biomarker for predicting clinical outcome of patients with AML.

## Results

### t(8;21) AML displays a distinct signature of aberrant DNA methylation

First, we used a DNA methylation microarray for 450 k CpG sites to profile genome-wide DNA methylation in AML blasts of patients with or without t(8;21) (Appendix Table S1) and normal bone marrow (NBM) CD34$^+$ cells. The heatmap displayed a unique epigenetic signature with global hypermethylation for t(8;21)$^+$ AML, which differed from either the t(8;21)$^-$ AML or NBM samples (Fig 1A). Unsupervised hierarchical clustering analysis revealed a clear segregation of t(8;21)$^+$ AML from t(8;21)$^-$ AML blasts or normal CD34$^+$ cells (Appendix Fig S1), despite the heterogeneity of AML blasts. It also showed that t(8;21)$^+$ cases were further divided into two sub-clusters (A and B) with distinguishable methylation profiles (Appendix Fig S1). Interestingly, sub-cluster A contained 5 of 7 t(8;21)$^+$ AML cases carrying Y chromosome deletion (Appendix Table S1).

Analysis of the differentially methylated probes within 10 kb from each gene transcriptional start site (TSS) identified a total of 888 differentially methylated probe sets (corresponding to 408 genes) in t(8;21)$^+$ AML [Fig 1B, $P < 0.001$ and methylation log ratio difference $> 1.5$ vs. t(8;21)$^-$ AML]. In t(8;21)$^+$ AML, functional pathway analysis indicated that 57 and 17 hypermethylated genes were grouped into the categories of cancer and AML, respectively (Fig 1C). Among these genes (Appendix Table S2), many are involved in tumour suppression [e.g. *RB1* (Dyson, 2016)], apoptosis [e.g. *BCL2L1* (Liu *et al*, 2015)], the cell cycle [e.g. *CCND1* (Mende *et al*, 2015), *CCND2* (Khanjyan *et al*, 2013)] or protein kinases [e.g. *KIT* (Wichmann *et al*, 2015), *GSK3-β* (Orme *et al*, 2016)]. By combined with 408 differentially methylated genes (Fig 1B), a systematic survey of the publicly available databases (Gardini *et al*, 2008; Zhang *et al*, 2015) identified nine transcriptional factor genes targeted by AML1-ETO that were differentially hypermethylated in t(8;21) AML (Fig 1D). The same approach was then used to analyse the data from TCGA methylome (Ley *et al*, 2013), which identified 13 genes differentially

hypermethylated in t(8;21) AML (Fig EV1). Of note, 4 of these 13 genes matched those identified in our distinct signature of t(8;21) AML (Fig 1D).

A strong hypermethylated signature of t(8;21) AML suggests a therapeutic benefit from hypomethylating agents (e.g. 5-azacytidine/5-Aza). Notably, 5-Aza treatment significantly increased apoptosis of AML1-ETO$^+$ AML cells (Appendix Fig S2), compared to their AML1-ETO$^-$ counterparts. Together, these findings indicate a distinct signature of aberrant DNA methylation in t(8;21) AML, consistent with the hypothesis that AML1-ETO alters epigenetic patterning to drive leukaemogenesis via deregulation of specific complementary gene sets, which may be susceptible to hypomethylating agents.

### THAP10 is epigenetically suppressed in t(8;21) leukaemia cells

Differential methylated region (DMR) analysis was then performed to explore the genes aberrantly methylated in t(8;21) AML. The top 10 genes with higher methylation levels in t(8;21)$^+$ than t(8;21)$^-$ AML and NBM are shown in Appendix Table S3. Notably, both sets of these top 10 genes and nine genes described above (Fig 1D) included *THAP10*, while the other nine genes listed in Appendix Table S3 were not found in the latter gene set. Moreover, methylation levels of all 14 probes in the *THAP10* promoter region were higher in AML1-ETO$^+$ than AML1-ETO$^-$ blasts or NBM cells (Appendix Fig S3), supporting *THAP10* as a novel target gene epigenetically regulated by AML1-ETO.

*THAP10* mRNA levels were then determined to validate the functional role of its promoter hypermethylation. Significantly, core-binding factor [CBF, including t(8;21)] AML cell lines displayed lower *THAP10* mRNA levels than non-CBF cells (Fig 2A). In a cohort of AML patients (Appendix Table S4), *THAP10* mRNA levels in AML blasts were lower than NBM cells, while the lowest level of *THAP10* was observed in t(8;21)$^+$ AML (Fig 2B). Virtually identical results were obtained from the analysis of data from 200 AML samples in TCGA database (Fig EV2; Ley *et al*, 2013). Interestingly, *THAP10* mRNA levels in bone marrow mononuclear cells of AML1-ETO$^+$ AML patients who achieved complete remission after induction chemotherapy were relatively higher than the same patients at relapse (Fig 2C). In an additional cohort of 124 AML/M2 patients (Appendix Table S5), *THAP10* mRNA levels were 0.9-fold lower in AML1-ETO$^+$ than AML1-ETO$^-$ cases (Fig 2D).

As *AML1-ETO* and *c-KIT* gene abnormalities are associated with poor outcome of patients with t(8;21) AML (Jiao *et al*, 2009; Qin *et al*, 2014), we examined the relationships among expression of *THAP10*, *AML1-ETO* and *c-KIT* in newly diagnosed t(8;21) AML patients (Appendix Table S5). Notably, *THAP10* mRNA level was inversely correlated with those of *AML1-ETO* (Fig 2E) and *c-KIT* (Fig 2F). Then, 76 patients with t(8;21) AML (Appendix Table S5) were grouped into quartiles according to *THAP10* levels and divided into *THAP10*-high ($n = 56$) and *THAP10*-low patients ($n = 20$; Appendix Fig S4). The results revealed that *THAP10*-high patients had significantly longer overall (Fig 2G), event-free (Fig 2H) or relapse-free (Fig 2I) survival than *THAP10*-low patients. Together, these findings indicate that *THAP10* gene silencing is associated with unfavourable outcomes of t(8;21) AML patients.

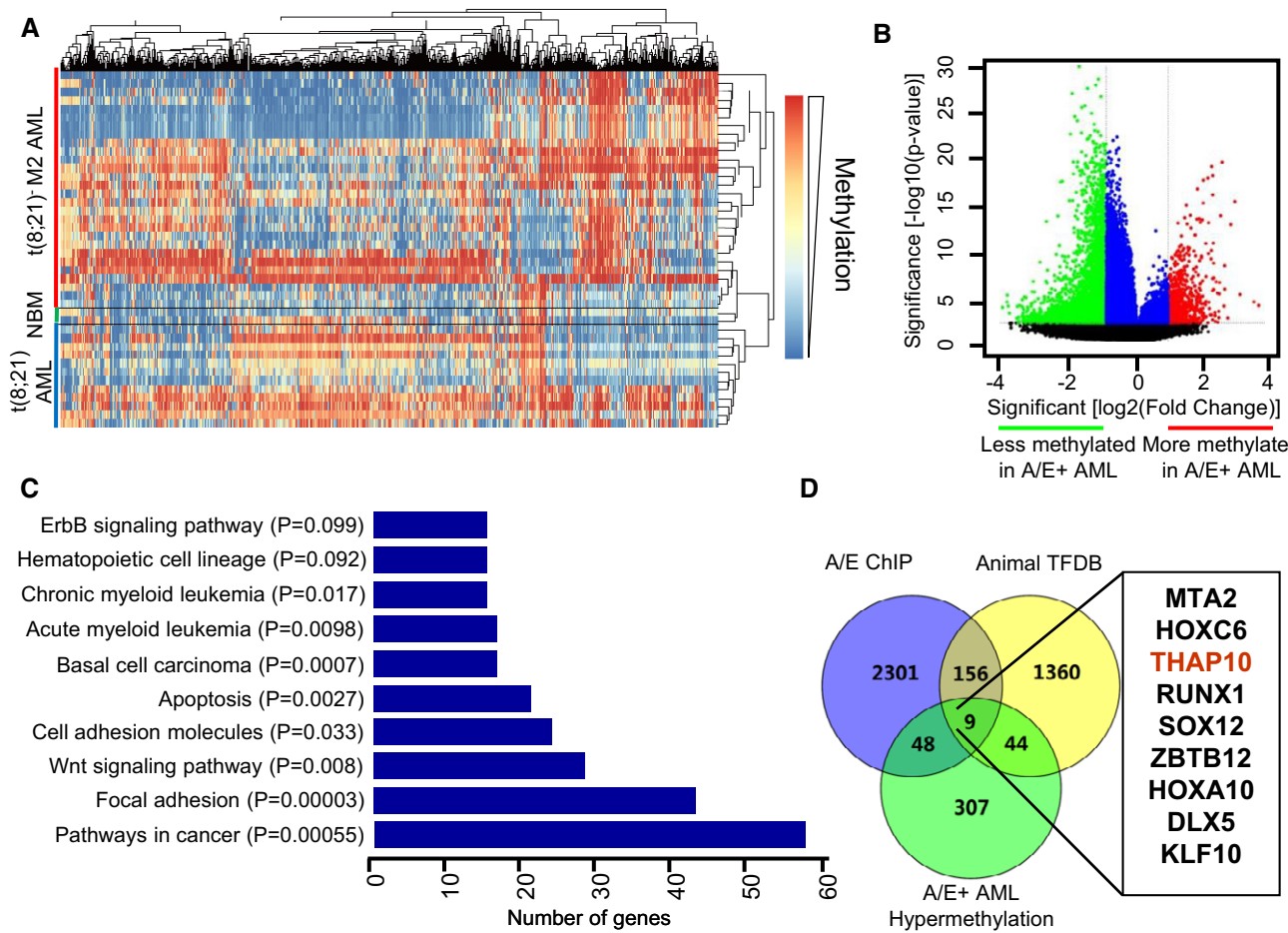

**Figure 1. AML1-ETO⁺ AML displays a unique genome-wide methylation profile compared to AML1-ETO⁻ AML and normal bone marrow CD34⁺ cells.**

A   Overview of two-way (genes against samples) hierarchical clustering of AML1-ETO⁺ (*n* = 12), AML1-ETO⁻ (*n* = 28) M2 AML blasts and normal bone marrow cells (NBM, *n* = 2) using the genes that vary the most among samples. Representative heatmap is shown to identify the aberrant DNA methylation signatures of specific genes in AML, compared to normal CD34⁺ haematopoietic cells obtained from healthy donors. In this heatmap, each column represents one gene, and each row represents one AML patient or healthy donor. Red bar, t(8;21)⁻ M2 AML. Blue bar, t(8;21)⁺ AML. Green bar, normal bone marrow (NBM).

B   Volcano plot illustrating the methylation difference between AML1-ETO⁺ (*n* = 12) and AML1-ETO⁻ (*n* = 28) AML samples, with the corresponding moderated *t*-test *P*-values. Red, probe sets that were significantly hypermethylated (*P* < 0.001, methylation difference > 1.5); green, probe sets that were significantly hypomethylated (*P* < 0.001, methylation difference < −1.5); blue, probe sets that did not have an absolute methylation difference > 1.5.

C   Functional enrichment analysis of differential DNA methylation genes (DMGs), based on the pathway enrichment analysis of hypergeometric distribution.

D   Venn diagrams for systematic survey of the publicly available database, by combining with 408 differentially methylated genes identified in the present study (see panel B), to identify the 9-gene set in t(8;21) AML.

Data information: All experiments were performed in triplicate.
Source data are available online for this figure.

## AML1-ETO epigenetically suppresses *THAP10* by binding to AML1-binding sites

AML1-ETO retains the binding capacity to AML1-binding sites at gene promoters, and it can also recruit additional transcriptional cofactors, mostly corepressors (Li *et al*, 2013b). Notably, *AML1-ETO* knock-down resulted in sharp increases (~15-fold compared to controls) in *THAP10* mRNA and protein levels in AML1-ETO⁺ SKNO-1 (Fig 3A). In contrast, ectopically expressing zinc-inducible HA-tagged *AML1-ETO* significantly reduced *THAP10* levels in U937 cells (Fig EV3A). However, despite the significant difference in mRNA levels of *THAP10*, there was only a moderate difference in

THAP10 protein levels between AML1-ETO⁻ HL60 and AML1-ETO⁺ Kasumi-1 cells (Fig 3A). This inconsistency suggests that other mechanism(s) may also contribute to the protein level of THAP10. Nevertheless, these results indicate that AML1-ETO inhibits *THAP10* expression.

A bioinformatics search (http://www.cbrc.jp/research/db/ TFSEARCH.html) at the 5′ end of the predicted "core promoter" sequence revealed seven putative AML1-binding sites surrounded by CpG islands in the *THAP10* upstream region (−2,000 nt to +1 nt relative to TSS; Fig 3B, upper panel). Accordingly, luciferase reporters containing a series of wild-type (P1–P8) or mutated (P4-M to P6-M) sequences of the *THAP10* promoter region (Fig 3B, lower panels)

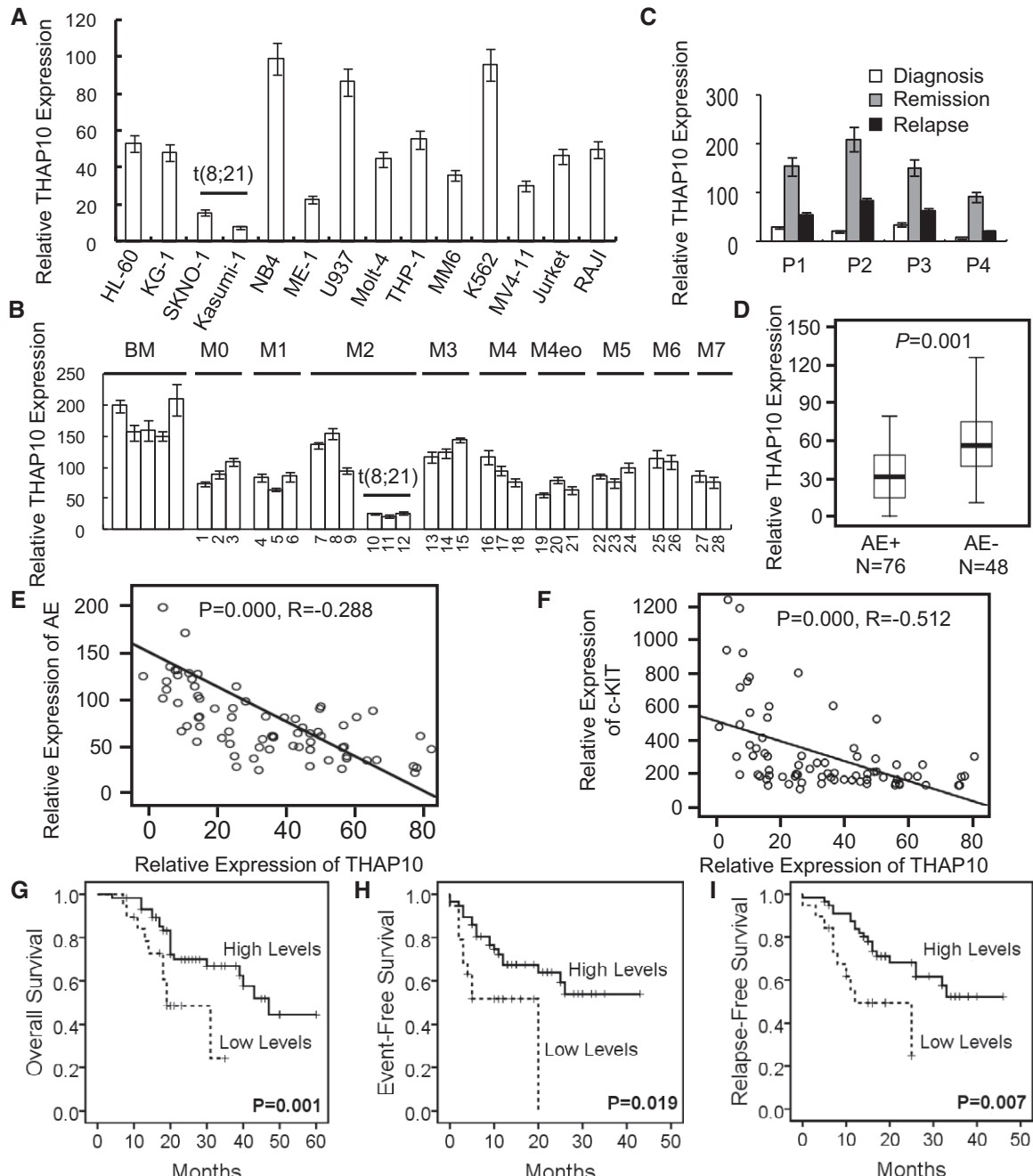

**Figure 2. THAP10 is down-regulated in t(8;21) AML and correlates with adverse clinical outcome.**

A, B   Relative qRT–PCR quantification of *THAP10* mRNA levels in the indicated leukaemia cell lines (A) and mononuclear cells (MNC) isolated from leukaemia patients (B) (n = 28).

C   Sequential analyses of *THAP10* mRNA levels in mononuclear cells isolated from bone marrow samples of four individual leukaemia patients at different stages of disease, including newly diagnosed, remission and relapse.

D   Comparison of *THAP10* mRNA levels in two categories of AML M2 subtype patients (total n = 124): AML1-ETO[+] (n = 76) and AML1-ETO[−] (n = 48) t(8;21) AML, P = 0.001. Data are represented as a boxplot, which shows the median value of mRNA levels (line in the middle of the box), the first and the third quartiles (lower and upper limits of each box, respectively). Wiskers display the highest and lowest data values.

E, F   Correlations in gene expression between *THAP10* and *AML1-ETO* (R = −0.288, P = 0.000, E) or *c-KIT* (R = −0.512, P = 0.000, F), the two-sided paired Pearson test was used for the significance of association.

G–I   Correlations of *THAP10* expression (high, n = 56; low, n = 20) with overall survival (P = 0.001, G), event-free survival (P = 0.046, H) and relapse-free survival (P = 0.039, I). The cut-off value (dash line) is $2 \times 10^1$ of *THAP10* level, by which high vs. low *THAP10* mRNA expression is defined, the log-rank test was used for the survival analysis.

Data information: Data are expressed as the mean ± SEM.
Source data are available online for this figure.

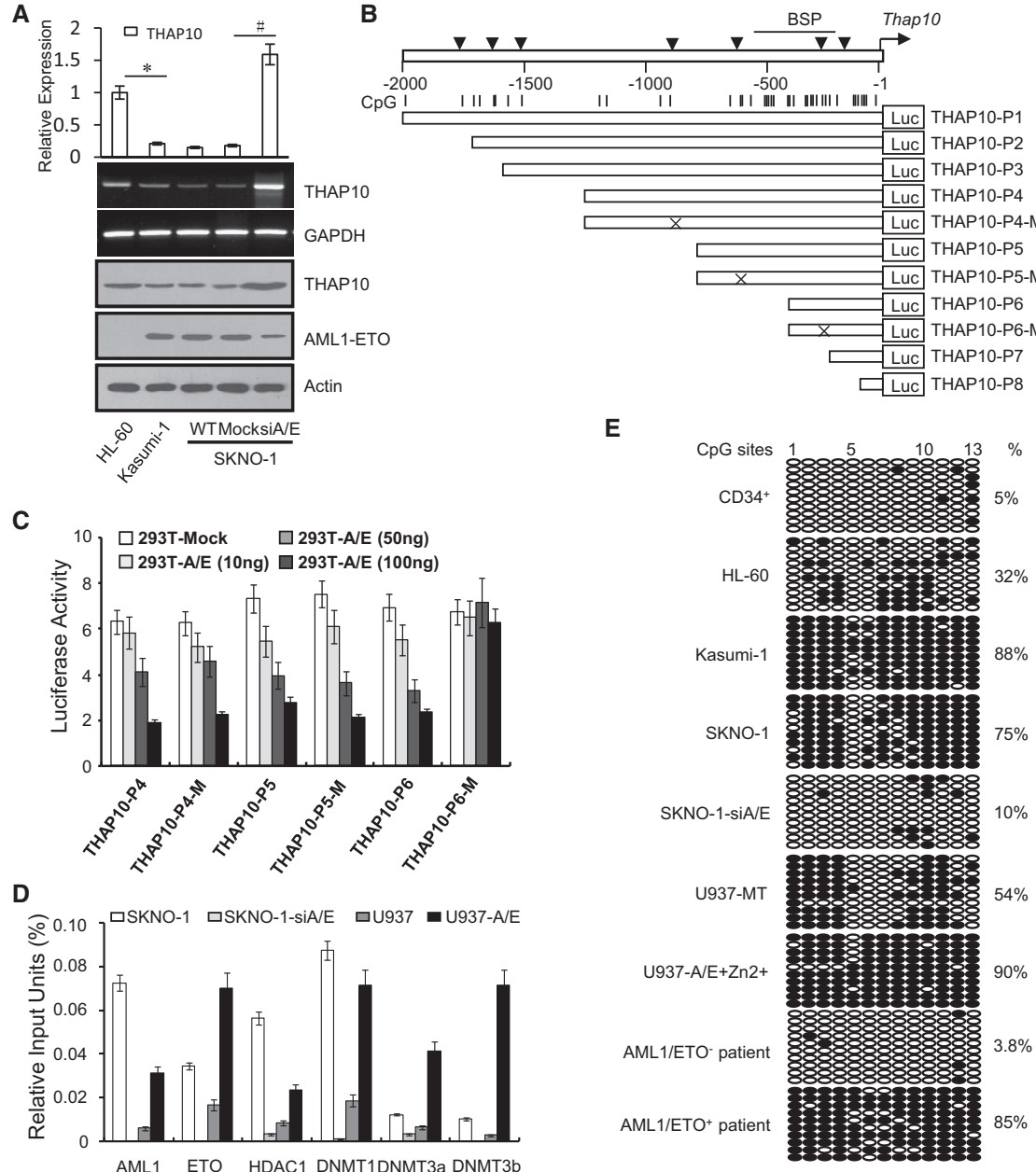

**Figure 3. *AML1-ETO* epigenetically suppresses *THAP10* via direct binding to its promoter region.**

A Upper panel: Relative quantification of *THAP10* mRNA levels in the indicated leukaemia cell lines. The results represent the mean of three independent evaluations ± SD (*$P$ = 0.001, #$P$ = 0.0007, Student's *t*-test was used for the comparisons). Middle and lower panels: mRNA and protein levels of *THAP10* and *AML1-ETO*, respectively.

B Upper panel: Schematic diagrams of the AML1-binding sites and the CpG islands along the *THAP10* genes. Numbers indicate the nucleotides relative to *pre-THAP10* (−1 nt). Vertical arrowheads indicate the AML1-binding sites; vertical lines indicate CpG dinucleotides; horizontal bar illustrates the regions analysed by bisulphite sequencing. Lower panels: A series of constructs containing different AML1-binding sites and their mutants.

C Luciferase reporter activities of human 293T cells transiently co-transfected for 48 h with luciferase reporters containing the wild-type sequence of the *THAP10* promoter or its mutant counterparts, together with increasing amounts of pcDNA3.0 vectors containing *AML1-ETO* or mock cDNA (293T-Mock). Triplicate values are defined as mean ± SD.

D Chromatin immunoprecipitation (ChIP) using the indicated antibodies or IgG, after which qRT–PCR was performed to evaluate the specificity of protein binding. Triplicate values are defined as mean ± SD.

E Genomic bisulphite sequencing to detect the methylation status of the DNA sequences surrounding the AML1-binding site (−280 nt) in the *pre-THAP10* gene upstream region in CD34[+] haematopoietic progenitors isolated from peripheral blood (PB) of healthy donors, the indicated leukaemia cell lines and primary leukaemic blasts. Each row of circles represents the sequence of an individual clone. Solid and empty circles represent methylated and unmethylated CpG dinucleotides, respectively.

Data information: All experiments were performed in triplicate.
Source data are available online for this figure.

were designed, and 293T cells were co-transfected with each reporter together with increasing amounts of *AML1-ETO* or empty vectors. Notably, ectopic *AML1-ETO* expression resulted in a dose-dependent decrease in luciferase activity of THAP10-P1-6 or THAP10-P4-M and THAP10-P5-M but not THAP10-P6-M and THAP10-P8, which lack an AML1-binding site (Figs 3C and EV3B), suggesting that AML1-ETO binds to the proximal AML1-binding sites of the *THAP10* promoter to suppress *THAP10*. Moreover, ChIP assays revealed that shRNA knock-down of AML1-ETO in SKNO-1 cells resulted in a sharp decrease in binding of AML1-ETO, HDAC1, DNMT1, DNMT3a and DNMT3b to the region around the AML1-binding sites of the *THAP10* promoter compared to control SKNO-1 cells (Fig 3D). In contrast, ectopic overexpression of AML1-ETO in U937 cells led to a marked increase in binding of these proteins compared to control U937 cells (Fig 3D). Similar results were obtained in primary AML1-ETO$^+$ vs. AML1-ETO$^-$ AML blasts (Fig EV3C).

Further, bisulphite sequencing assays revealed a higher frequency of methylated CpG dinucleotides encompassing *THAP10* in AML1-ETO$^+$ SKNO-1 (75%) and U937-A/E cells (90%) than U937 (54%) and HL-60 (32%) or normal CD34$^+$ cells (5%) (Fig 3E). In contrast, *AML1-ETO* knock-down in SKNO-1 cells dramatically diminished *THAP10* CpG methylation (Fig 3E). The frequency of methylated CpG dinucleotides encompassing endogenous *THAP10* was higher in AML1-ETO$^+$ (85%) than AML1-ETO$^-$ (3.8%) AML blasts (Fig 3E). Interestingly, while treatment with 5-Aza increased *THAP10* expression by ~4.0-fold in AML1-ETO$^+$ cells displaying strong hypermethylation of *THAP10* (Fig 3E), co-treatment with 5-Aza and the HDAC inhibitor TSA further increased its expression (Fig EV3D). These findings are likely due to increasing acetylation (activation) of the *THAP10* promoter via inhibition of the co-repressor HDAC1, which was recruited to the AML1-binding sites of *THAP10* (Fig 3D) by HDAC inhibitors (e.g. TSA). Treatment with another hypomethylating agent (decitabine) +/− the clinically relevant HDAC inhibitor chidamide also increased *THAP10* expression *ex vivo* in primary blasts (Fig EV3E) as well as *in vivo* in patients enrolled in an ongoing clinical trial (NCT02886559; Fig EV3F). Together, these findings suggest that AML1-ETO in an abnormal chromatin remodelling complex with other cofactors (e.g. DNMTs) mediates hypermethylation of CpG islands around AML1-binding sites on the *THAP10* promoter, while probably increasing histone deacetylation (e.g. by HDAC1) on the *THAP10* promoter as well, which might represent a novel mechanism responsible for suppression of *THAP10* expression in t(8;21) AML.

### THAP10 is a *bona fide* target of *miR-383* that is up-regulated in AML1-ETO$^+$ leukaemia cells

In addition to the mechanism described above in which AML1-ETO directly suppresses the THAP10 gene by methylating its promoter, we then examined whether other mechanism(s) were also involved in suppression of THAP10 expression in AML-ETO$^+$ AML. MicroRNAs (miRNAs) play an important role in the lineage differentiation of haematopoietic cells by regulating expression of oncogenes or tumour suppressors (Vian *et al*, 2014). Deregulation of miRNA expression has been shown to be involved in multistep carcinogenesis and has rapidly emerged as a novel therapeutic target (Li *et al*, 2013b; Dorrance *et al*, 2015; Tarighat *et al*, 2016). Computational analysis for miRNA target prediction using the TargetScan

algorithm (http://www.targetscan.org) revealed *THAP10* as a putative target of *miR-383* (Fig 4A). To validate the functional role of *miR-383* in regulation of *THAP10* expression, 293T cells were co-transfected with *THAP10* 3′-UTR luciferase reporters and *pre-miR-383*. As shown in Fig EV4A, expression of *miR-383* resulted in a ~50% reduction in luciferase activity of wild-type *THAP10* 3′-UTR but not in *miR-383*-binding site-mutated *THAP10* 3′-UTR (Fig 4A). Furthermore, administration of synthetic *miR-383* also reduced endogenous THAP10 expression at the mRNA level in HL-60 and NB4 cells (Fig EV4B) and the protein level (Fig 4B). Taken together, these results indicate that *THAP10* is a *bona fide* target of *miR-383*.

qRT–PCR analysis revealed higher *miR-383* levels in t(8;21)$^+$ than t(8;21)$^-$ AML cell lines (Fig 4C) and blasts (Fig 4D) in a cohort of AML patients (Appendix Table S4). Interestingly, *miR-383* expression was decreased in bone marrow mononuclear cells of AML1-ETO$^+$ patients who achieved complete remission following induction chemotherapy, compared to the same patients at relapse (Fig 4E). These findings were further validated by results showing that *miR-383* levels were twofold higher in AML1-ETO$^+$ than AML1-ETO$^-$ AML blasts (Fig 4F) in an additional cohort of 124 AML cases (Appendix Table S5).

Furthermore, *AML1-ETO* knock-down decreased *miR-383* expression by ~sevenfold in AML1-ETO$^+$ cells, compared to controls (Fig EV4C). In contrast, ectopic expression of *AML1-ETO* in U937 cells increased *miR-383* levels by 2.4-fold (Fig EV4D). Consistently, qRT–PCR analysis revealed an inverse correlation between *miR-383* and *THAP10* expression (Fig 4G), but a positive correlation between *miR-383* and *AML1-ETO* expression (Fig 4H), in a cohort of newly diagnosed t(8;21) AML patients (Appendix Table S5). Together, these findings suggest that *miR-383* expression correlates negatively with *THAP10* but positively with *AML1-ETO* in t(8;21) AML.

### AML1-ETO transcriptionally induces expression of *miR-383*

The NHR1 domain of AML1-ETO provides a docking site for p300 binding, allowing it to co-localize at the regulatory regions of many AML1-ETO target genes (Wang *et al*, 2011). A bioinformatics search (http://www.cbrc.jp/research/db/TFSEARCH.html) of the *miR-383* at the 5′ end of the predicted "core promoter" sequence revealed three putative AML1-binding sites and one p300-binding site (Fig 5A, upper panel). Accordingly, luciferase reporters containing a series of wild-type (MIR-P1 to MIR-P4) or mutated (MIR-P1-M to MIR-P3-M) sequences of the *miR-383* promoter region (Fig 5A, lower panels) were designed, and 293T cells were co-transfected with each reporter together with increasing amounts of *AML1-ETO* or empty vectors. Ectopic *AML1-ETO* expression resulted in a dose-dependent increase in luciferase activity of MIR-P1, MIR-P1-M and MIR-P2, but not MIR-P2-M, MIR-P3 and MIR-P4 (Fig 5B), suggesting that the AML1-binding sites in the promoter region (−676 nt relative to +1 nt) of *miR-383*, which is close to the p300-binding site (−669 nt relative to +1), might be involved in *AML1-ETO*-mediated expression of *miR-383*.

ChIP analysis revealed the presence of AML1-ETO and p300, rather than HDAC1, DNMT1, DNMT3a and DNMT3b, in the *miR-383* promoter region surrounding the AML1-binding sites (Fig 5C). Furthermore, ChIP analysis using antibody against acetylated histone H3 showed that acetylation levels within the *miR-383* promoter region around the AML1-ETO- and p300-binding sites

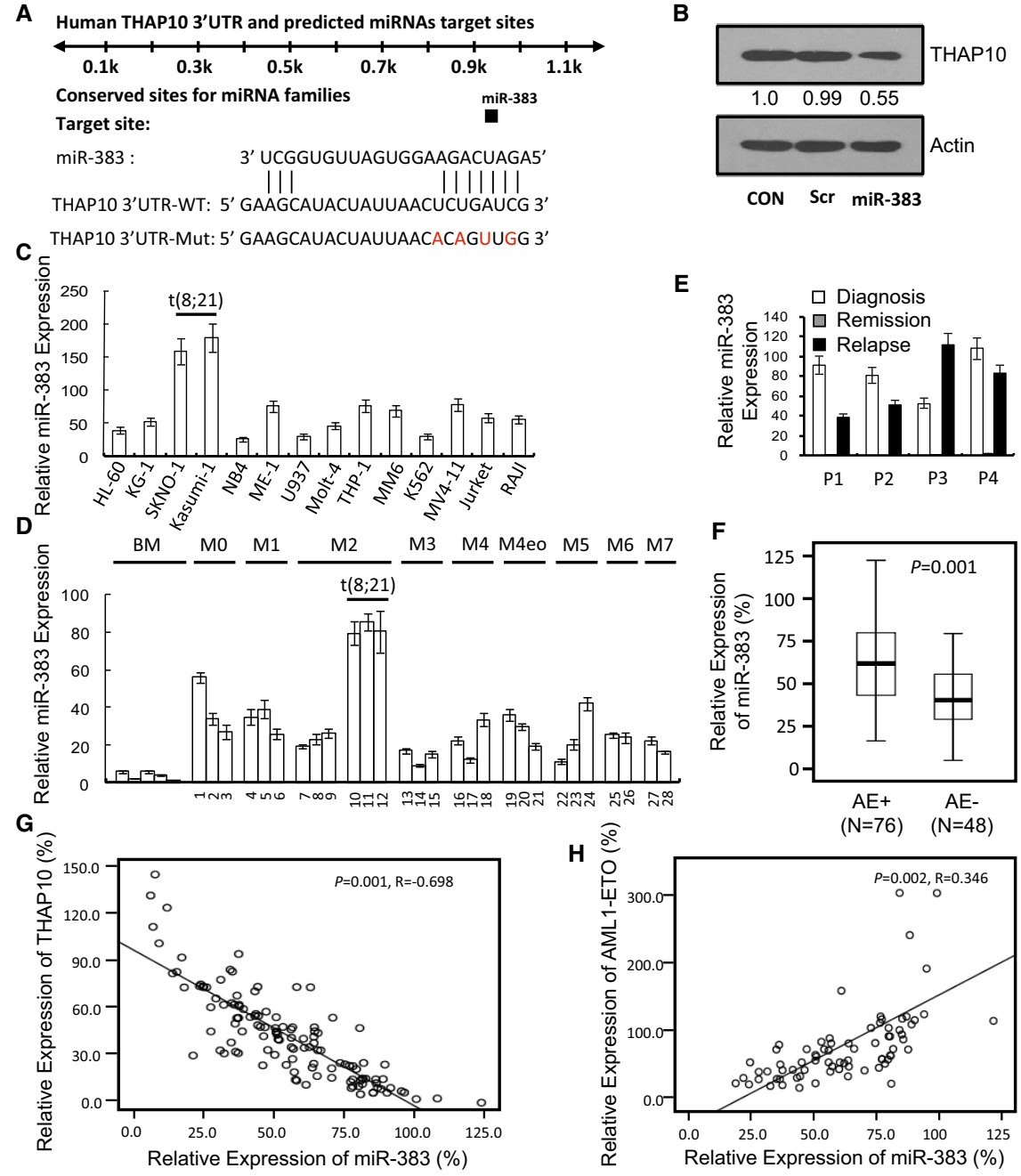

**Figure 4. *miR-383* negatively correlates with *THAP10* levels but positively correlates with *AML1-ETO*.**

A The sequences indicating the putative interaction site between the *miR-383* seed region (nucleotides 2–8) and the 3′-UTRs of *THAP10* wild-type (WT) or mutated derivatives.

B Western blot analysis monitoring the effect of synthetic *miR-383* on THAP10 protein levels.

C, D Relative qRT–PCR quantification of *THAP10* mRNA levels in the indicated leukaemia cell lines (C) and mononuclear cells (MNC) isolated from leukaemia patients (D) (*n* = 28). Triplicate values are defined as mean ± SD.

E Sequential analyses of *miR-383* levels in mononuclear cells isolated from bone marrow samples of four individual leukaemia patients at the newly diagnosed, remission and relapse stages, respectively. Triplicate values are defined as mean ± SD.

F Relative qRT–PCR quantification of *THAP10* mRNA levels in two groups of AML M2 subtype patients (total *n* = 124): AML1-ETO[+] and AML1-ETO[−] t(8;21) AML (*P* = 0.001). Data are represented as a boxplot, which shows the median value of mRNA levels (line in the middle of the box), the first and third quartiles (lower and upper limits of each box, respectively). Whiskers display the highest and lowest data values.

G, H Correlation of *miR-383* expression with *THAP10* (*R* = −0.689, *P* = 0.001, G) or *AML1-ETO* levels (*R* = 0.346, *P* = 0.002, H). Two-sided paired Pearson test was used for the significance of association.

Data information: All experiments were performed in triplicate.
Source data are available online for this figure.

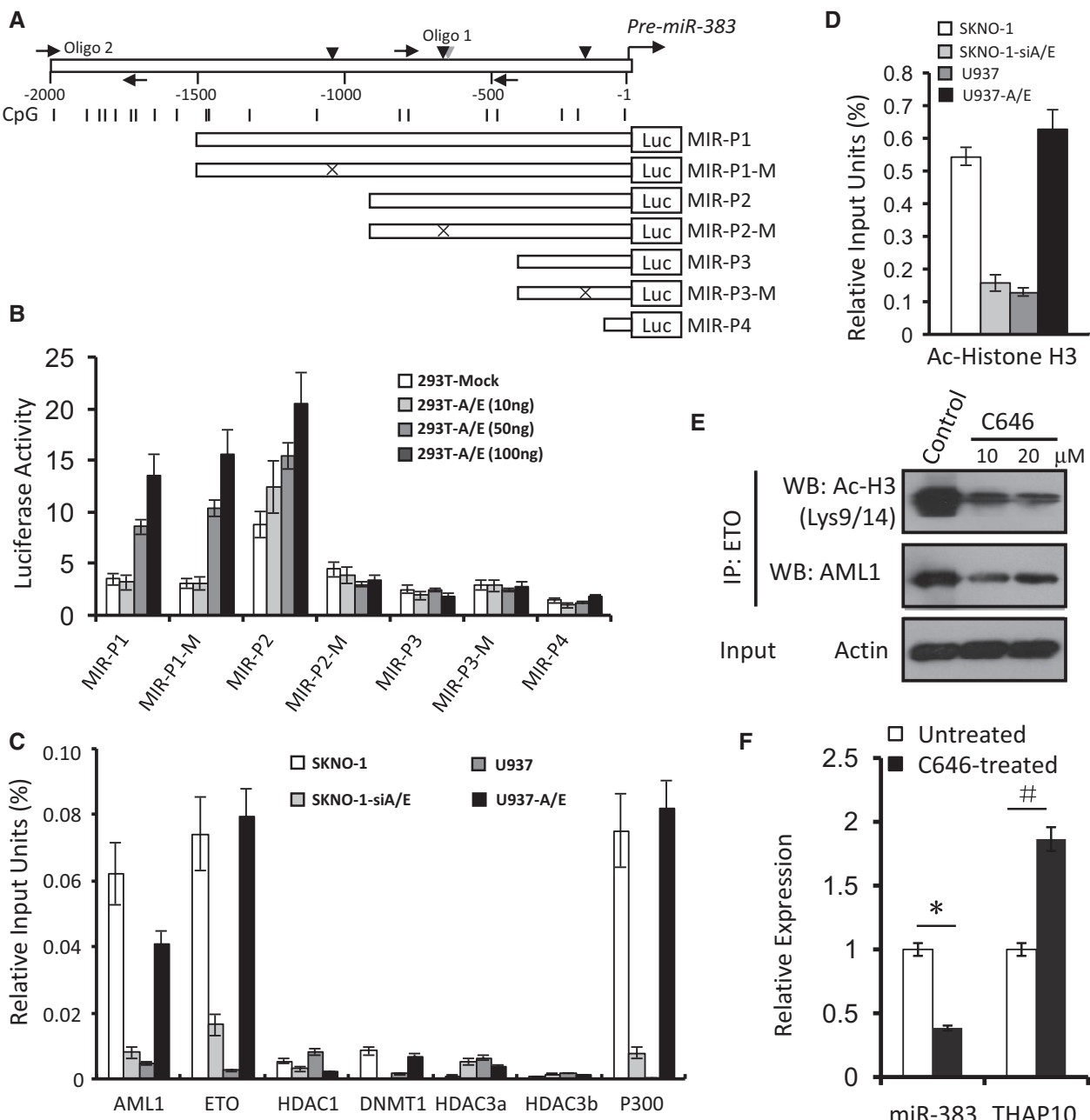

**Figure 5. AML1-ETO epigenetically activates expression of *miR-383*.**

A Upper panel: Schematic diagrams of the AML1- and p300-binding sites and the CpG islands along the *pre-miR-383* genes. Numbers indicate the nucleotides relative to *pre-miR-383* (+1 nt). Vertical arrowheads indicate AML1 (dark) and p300 (grey) binding sites; horizontal arrows indicate the location of the primers used for the ChIP assays; vertical lines indicate CpG dinucleotides. Lower panels: A series of constructs containing different AML1-binding sites and their mutants.

B Luciferase reporter activities of human 293T cells transiently co-transfected for 48 h with luciferase reporter constructs containing the wild-type sequence of the *miR-383* regulatory regions or its mutant counterparts, together with increasing amounts of pcDNA3.0 containing *AML1-ETO* or mock cDNA (293T-Mock). Triplicate values are defined as mean ± SD.

C ChIP using the indicated antibodies or IgG, after which qRT–PCR was performed to evaluate the specificity of protein binding. Triplicate values are defined as mean ± SD.

D ChIP using Ac-histone H3 antibody or IgG, after which qRT–PCR was performed for amplification of the region in *miR-383* containing the predicted AML1-binding site in the upstream region of *miR-383* with the predicted p300-binding site, to evaluate the specificity of protein binding. Triplicate values are defined as mean ± SD.

E Kasumi-1 cells were treated with C646 (10 or 20 μM) for 24 h, after which acetylation of histone H3 was analysed by immunoprecipitation (IP) and Western blot.

F qRT–PCR quantification of *miR-383* and *THAP10* mRNA levels in the Kasumi-1 cells treated with or without C646 (10 μM) for 24 h. The results represent the mean of three independent evaluations ± SD (*P = 0.0001, #P = 0.002). Student's *t*-test was used for the comparisons.

Data information: All experiments were performed in triplicate.
Source data are available online for this figure.

were higher in AML1-ETO⁺ than AML1-ETO⁻ cells (Fig 5D). Given the role of the interaction between AML1-ETO and p300 in regulation of histone acetylation (Wang *et al*, 2011), the chemical p300 inhibitor C646 was then used to determine the function of p300 in acetylation of the *miR-383* promoter region around the AML1-ETO- and p300-binding sites. Indeed, C646 treatment substantially decreased acetylation of histone H3 (lysine 9/14) as well as AML1-ETO in Kasumi-1 cells (Fig 5E), while it increased *THAP10* levels (Fig 5F). Together, these findings indicate that AML-ETO recruits p300 to the AML1-binding sites in the *pre-miR-383* regulatory region and therefore transcriptionally activates *miR-383* expression, which in turn suppresses *THAP10* expression, suggesting an alternative indirect mechanism for AML-ETO to negatively regulate THAP10 expression in t(8;21) AML.

### THAP10 is a nuclear protein that inhibits proliferation but promotes differentiation of t(8;21) AML cells

In *Drosophila*, several THAP proteins are putative transcription factors (Sabogal *et al*, 2010), indicating that human THAP10 may also regulate transcription. Thus, we first examined whether THAP10 was localized in the nucleus, where transcriptional factors function to regulate gene expression. To this end, NIH3T3 cells were transfected with EGFP-fused *THAP10* cDNA to monitor subcellular localization of THAP10 protein. In contrast to GFP, which was distributed throughout the cell, GFP-THAP10 was found only in the nucleus (Fig 6A). Western blot analysis further validated nuclear localization of endogenous THAP10 protein in Kasumi-1 cells transduced with lentivirus vector expressing *THAP10* (Fig EV4E). Moreover, immunofluorescence staining for THAP10 also revealed that the endogenous THAP10 protein predominantly localizes in the nuclei of HL-60 cells and, to a lesser extent, in Kasumi-1 cells and t(8;21) AML blasts (Fig EV4F). Together, these findings suggest that the *THAP10* gene encodes a nuclear protein.

To examine the functional role of *THAP10* in leukaemogenesis of t(8;21) AML, microarray analysis of global gene expression was performed to identify the genes and pathways that correlate with *THAP10* (Fig 6B) in Kasumi-1 cells stably expressing *THAP10* (Fig EV4G). Strikingly, classification based on the gene ontology (GO) annotation showed that in *THAP10*-expressing Kasumi-1 cells, the significantly down-regulated genes were related to either receptor tyrosine kinase signalling pathways or cell differentiation (Appendix Table S6), suggesting the selectivity of *THAP10* in inhibition of gene expression. These results were further validated by qPCR using specific primers for 15 distinct genes that were selected either from the transmembrane receptor protein tyrosine kinase signalling pathway or from the haematopoietic progenitor cell differentiation pathway. All these 15 genes were down-regulated by at least twofold in *THAP10*-expressing Kasumi-1 cells compared to controls (Fig 6C). For detection of the total tyrosine phosphorylation, Western blot analyses showed a diminution of approximately 50% in p-Tyr levels in *THAP10*-expressing Kasumi-1 cells compared with mock-transduced cells (Appendix Fig S5). Thus, these findings suggest that *THAP10* inhibits cell proliferation by suppressing the genes associated with the enzyme-linked receptor signalling cascade, which may promote cell differentiation.

The role of *THAP10* in myeloid cell proliferation/differentiation was then examined. Ectopic *THAP10* expression significantly reduced proliferation of Kasumi-1 cells (Fig 6D). Conversely, shRNA knock-down of *THAP10* (Fig EV5A) markedly promoted cell proliferation (Fig 6E) in HL-60 cells that displayed a relatively high basal level of THAP10 (Fig 3A). Moreover, ectopic *THAP10* expression led to significant reductions in both number (~45%) and size of CFUs in Kasumi-1 cells (Fig 6F), while *THAP10* shRNA knock-down increased the number (~75%) and size of CFUs in HL-60 cells (Fig 6G). Finally, ectopic *THAP10* expression also induced myelomonocytic differentiation of SKNO-1 cells, as shown by morphological changes indicating maturation towards granulocytes (e.g. chromatin condensation, decreased nuclear/cytoplasmic ratio and appearance of primary granules; Fig 6H), as well as an increase (1–17.5%) in CD11b⁺ differentiated cells (Fig 6I). Taken together, these findings indicated that the nuclear protein THAP10 inhibits proliferation but promotes differentiation of t(8;21) AML cells.

### The *AML1-ETO/miR-383/THAP10* axis plays a functional role in leukaemogenesis of t(8;21) AML *in vivo*

Finally, the anti-leukaemic function of *THAP10* was examined *in vivo*. Kasumi-1 cells ectopically expressing *THAP10* were engrafted subcutaneously in the flank of immunocompromised BALB/c nude mice. Tumours formed from *THAP10*-expressing cells were significantly smaller than the empty-vector control ($P = 0.0016$) or parental counterparts ($P = 0.0015$) at day 28 post-cell inoculation (Fig 7A–C), with an average tumour weight of 1.80 g for *THAP10*-expressing cells vs. 5.25 g for empty-vector controls ($P = 0.0009$, Fig 7D). Ectopic *THAP10* expression markedly increased *THAP10* mRNA (Fig EV5B, upper panel) and protein levels (Fig EV5B, lower panels) in xenografts.

Similar experiments were performed to examine the effect of *miR-383* on tumour growth. t(8;21)⁺ SKNO-1 cells were inoculated subcutaneously into the flank of BALB/c nude mice as described above. When tumour size reached 50 mm³, synthetic anti-*miR-383* or scramble oligonucleotides were intratumourally injected as illustrated in Fig EV5C. On day 14 post-first treatment, treatment with anti-*miR-383* significantly reduced tumour size ($P = 0.0025$ vs. scramble oligonucleotide; $P = 0.0018$ vs. untreated control; Fig 7E–G), with an average tumour weight of 2.10 g for the anti-*miR-383* vs. 5.15 g for the scramble oligonucleotide groups ($P = 0.001$, Fig 7H). Finally, anti-*miR-383* treatment markedly diminished *miR-383* expression (Fig EV5D, upper panel), while it increased THAP10 protein levels (Fig EV5D, lower panels) in tumour tissue. To determine whether the *in vivo* activity of anti-*miR-383* is selective against AML-ETO⁺ AML, t(8;21)⁻ HL-60 cells were inoculated subcutaneously and intratumourally injected with anti-*miR-383* or scramble oligonucleotides as described above. As shown in Fig EV5E and F, there was no significant difference in either tumour size or weight between anti-*miR-383* and scramble oligonucleotide (4.85 g vs. 4.63 g, $P = 0.700$). Collectively, these findings strongly indicate that *THAP10* expression, which is negatively regulated by *miR-383*, functionally inhibits leukaemogenesis of t(8;21) AML.

## Discussion

Aberrant DNA methylation and/or deacetylation are major factors in epigenetic gene silencing. Indeed, disruption of DNA methylation

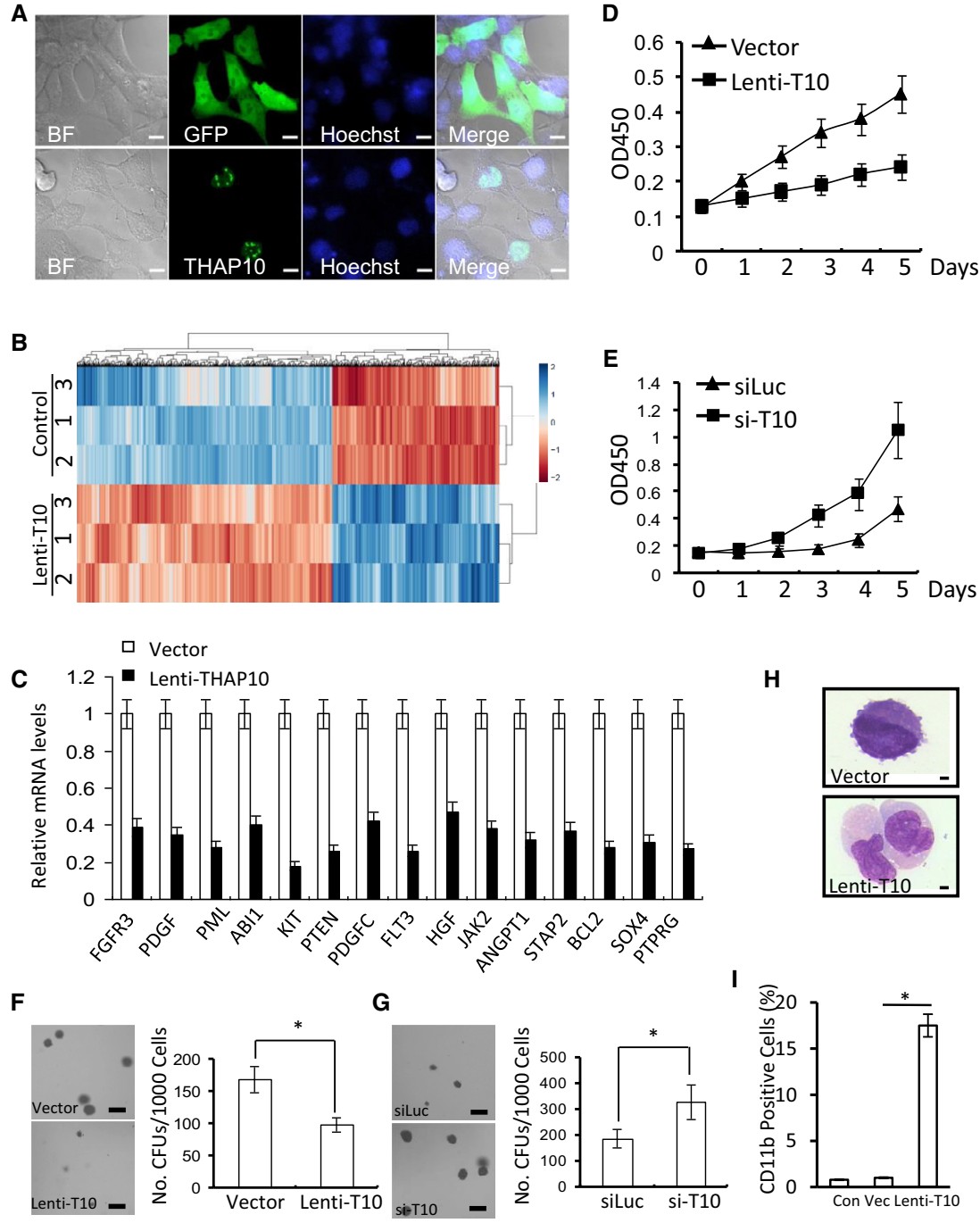

**Figure 6. THAP10 localizes in the nuclei and inhibits proliferation but promotes differentiation of t(8;21) AML cells.**

A    Nuclear localization of THAP10 in NIH3T3 cells. Scale bar, 10 μm.

B    Representative heatmap of the aberrant gene profile of Kasumi-1 cells transduced with or without *THAP10* using lentivirus. Each column indicates a gene, and 1–3 indicate three independent pools of cells.

C    Fold changes in the mRNA levels for 15 selected genes measured by RT–PCR.

D    Growth curve of Kasumi-1 cells transduced with *THAP10* (Lenti-T10) or lentiviral empty vector (Vector).

E    Growth curve of HL-60 cells transfected with synthetic si-*THAP10* (si-T10) or scramble (siLuc) sequences.

F, G  Analysis of CFUs in Kasumi-1 (F, *P = 0.026) and HL-60 cells (G, *P = 0.018) using a methylcellulose culture system. Scale bar, 1 mm. Two-sided Student's *t*-test was used for the comparisons.

H    Morphological analysis of Kasumi-1 cells at 4 days after transduction with empty vector or Lenti-THAP10 (T10). Scale bar, 10 μm.

I    Flow cytometric analysis of CD11b expression in Kasumi-1 cells at 144 h after transduction with Lenti-THAP10 (T10) or control vector (Vec). Wild-type Kasumi-1 cells as control (Con) (*P = 0.005). Two-sided Student's *t*-test was used for the comparisons.

Data information: All experiments were performed in triplicate. In (C–G and I), the results represent the mean of 3 independent evaluations ± SD.

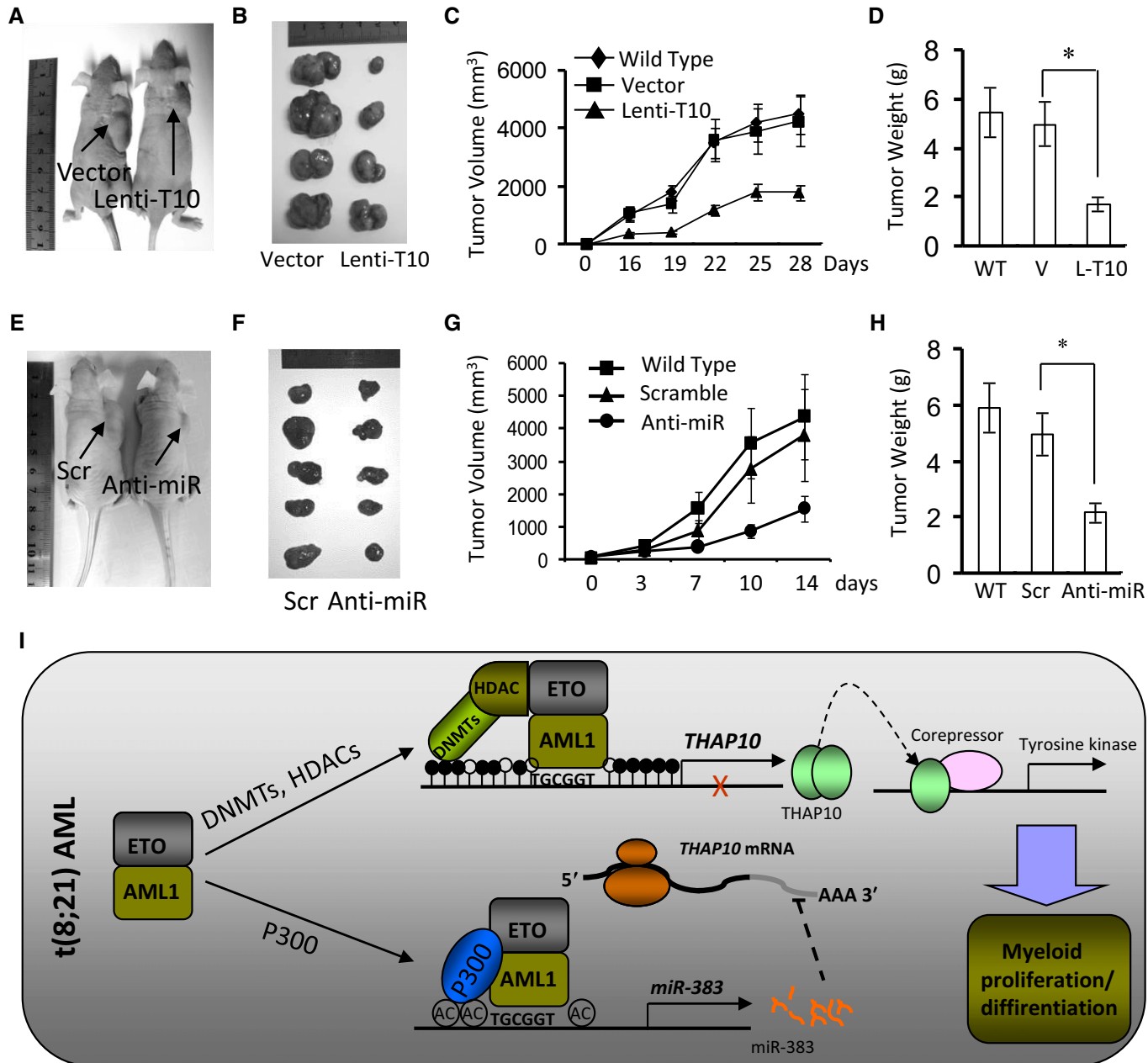

**Figure 7.  *THAP10* expression or *miR-383* inhibition suppresses tumour growth of t(8;21) AML cells *in vivo*.**

A   Representative photographs of mice in the lentivirus empty vector and Lenti-*THAP10* (T10) groups captured at the end of the experiment (day 28).

B   Tumour growth in mice engrafted with Kasumi-1 cells ectopically expressing *THAP10*.

C   Tumour volumes measured at the indicated days during the experiment (*n* = 6 for each group of wild type, empty vector and Lenti-T10). Values are defined as mean ± SD.

D   The mean ± SD of tumour weight at day 28 for the indicated groups (WT, wild type; V, empty vector; L-T10, Lenti-THAP10; *n* = 6 for each group; *P = 0.006). Two-sided Student's *t*-test was used for the comparisons.

E   Representative photographs of mice in the scramble and synthetic anti-*miR-383* treated groups, captured at the end of the experiment (day 14).

F   Tumour growth in mice engrafted with SKNO-1 cells ectopically expressing anti-*miR-383*.

G   Tumour volumes measured at the indicated days during the experiment (wild type, *n* = 5; scramble, *n* = 6; synthetic anti-*miR-383*, *n* = 6). Values are defined as mean ± SD.

H   The mean ± SD of tumour weight at the end of the experiment (WT, wild type, *n* = 5; Scr, scramble, *n* = 6; Anti-miR, anti-*miR-383*, *n* = 6; *P = 0.009). Two-sided Student's *t*-test was used for the comparisons.

I   A schematic model for the epigenetic suppression of *THAP10* by AML1-ETO directly or indirectly via *miR-383* and the interplays between AML1-ETO and the co-repressors DNMTs and HDACs or the co-activator p300. Epigenetic suppression of *THAP10* represents a mechanistic link between AML1-ETO fusion protein and the tyrosine kinase cascades by targeting their downstream genes, which mediate inhibition of myeloid proliferation and promotion of cell differentiation in t(8;21) AML.

Data information: All experiments were performed in triplicate. In (D and H), data are expressed as the mean ± SD.

patterns is frequently present in aberrant development and leukaemogenesis (Oakes *et al*, 2016). The present study unveiled a novel epigenetic mini-circuitry of *AML1-ETO/THAP10/miR-383* in t(8;21) AML, which plays a functional role in leukaemogenesis of this poor-prognostic AML subtype. The oncogenic transcription factor AML1-ETO, a fusion protein generated by t(8;21) translocation, can recruit DNMTs to and subsequently hypermethylate the promoters of its target genes (Fazi *et al*, 2007a; Li *et al*, 2013b). In this study, we showed that t(8;21) AML blasts displayed a specific methylation signature, distinct from those of t(8;21)$^-$ AML and normal CD34$^+$ cells, which was functionally associated with AML1-ETO.

We identified *THAP10* as a target gene of AML1-ETO. The oncogenic effect of AML1-ETO is primarily linked to its ability to form oligomeric complexes with an increased affinity for HDACs and DNMTs, which render AML1-ETO a potent transcriptional repressor of AML1-target genes. In this context, it has been reported that RARβ2, a retinoic acid (RA)-regulated tumour suppressor gene, is silenced by aberrant DNA methylation in a variety of human malignancies, including t(8;21) AML (Fazi *et al*, 2007b). Here, we showed that AML1-ETO directly bound to and hypermethylated the promoter region of *THAP10*, resulting in epigenetic suppression of this gene. *THAP10* and *LRRC49* (leucine rich repeat containing 49), both located on chromosome 15q23 in close proximity, were simultaneously suppressed due to hypermethylation of the bidirectional promoter region in breast cancer (De Souza Santos *et al*, 2008). The present findings indicate that *THAP10* acts as a tumour suppressor in t(8;21) AML, suggesting that AML1-ETO promotes leukaemogenesis via epigenetic suppression of *THAP10*. Therefore, *THAP10* expression may predict a favourable clinical outcome of patients with t(8;21) AML. The relevance of this finding in AMLs is underlined by the fact that aberrant heterochromatic gene silencing can represent an alternative mechanism to gene mutation or deletion for the transcriptional repression of tumour suppressor genes (Baylin & Ohm, 2006; Mack *et al*, 2015). However, we could not rule out the possibility that other AML1-ETO-targeted genes (e.g. those depicted in Fig 1D) might also be involved in leukaemogenesis driven by this oncogenic fusion protein.

In addition to direct epigenetic suppression of the THAP10 gene via methylation of its promoter by AML1-ETO, a question then arose whether this represents the sole mechanism for AML1-ETO to inhibit THAP10 expression or whether other mechanism(s) could also be involved in suppression of THAP10 expression in AML-ETO$^+$ AML. We further demonstrated that *THAP10* is also a *bona fide* target of *miR-383*, a miRNA associated with several types of solid tumour (Lian *et al*, 2010; He *et al*, 2013; Li *et al*, 2013a; Riaz *et al*, 2013; Zhao *et al*, 2014). Interestingly, AML1-ETO bound to the promoter of *miR-383* and activated its expression, while *miR-383* in turn suppressed expression of *THAP10*. Therefore, these findings indicate that *AML1-ETO* inhibits expression of the tumour suppressor THAP10 directly via epigenetic suppression of the *THAP10* promoter and indirectly through transcriptional activation of *miR-383* in t(8;21) AML (Fig 7I). This hypothesis is supported by findings indicating that the differentiation response to increased THAP10 can be restored in t(8;21) cell lines and blasts by (i) changing the methylation status at regulatory sites on the AML1-ETO target gene *THAP10* by AML1-ETO knock-down via RNAi or pharmacologic treatment with the demethylating agent 5-azacytidine and (ii) impairing the

interaction between AML1-ETO and the transcriptional co-activator p300 using the chemical inhibitor C646 or treatment with synthetic anti-*miR-383*.

Recent evidence suggests that alterations in transcription factors and tyrosine kinases represent two classes of the most frequently detected genetic events in human leukaemias. However, whether a causal relationship exists between the two genetic events remains largely unknown. The large-scale gene expression profiling in the present study showed that THAP10 acts as a transcriptional repressor to selectively modulate the enzyme-linked receptor tyrosine kinase signalling cascade and cell differentiation in t(8;21) AML. Thus, epigenetic suppression of *THAP10* may link to both the first hit (e.g. AML1-ETO) and the second hit (e.g. tyrosine kinase signalling cascade). Thus, we hypothesize that "loss of function" of the transcriptional repressor *THAP10* may result in activation of a key tyrosine kinase pathway, which then promotes cell proliferation but blocks cell differentiation of myeloid progenitors. Indeed, ectopic expression of THAP10 resulted in a clear increase in differentiated cells with specific morphologic features and the CD11b marker. However, the mechanism(s) underlying THAP10-mediated cell differentiation remains to be defined. Interestingly, there were significant inverse correlations between *THAP10* and *AML1-ETO* or tyrosine kinase *C-KIT* mRNA levels, implying that high levels of *THAP10* may indicate a good clinical outcome of patients with t(8;21) AML. Therefore, these findings strongly suggest that THAP10 is a tumour suppressor candidate as well as a potential biomarker for leukaemogenesis of t(8;21) AML.

In summary, this study identified *AML1-ETO/THAP10/miR-383* as a novel epigenetic mini-circuitry in t(8;21) AML, thereby providing new insight into the mechanism of action of *AML1-ETO* in driving leukaemogenesis. These findings also suggest that *THAP10* and *miR-383* are novel biomarkers and therapeutic targets in this AML subtype. Furthermore, this study presents new evidence for a functional link between the AML1-ETO protein, cell proliferation and differentiation, and the abnormal tyrosine kinase signalling pathway in human t(8;21) AML. Our study also provided new insight into the molecular processes required for myeloid differentiation and identified possible new targets for the development of novel therapeutic approaches to leukaemia. Future studies should aim to (i) determine the roles of other human THAP proteins in cell proliferation, (ii) define the composition of the endogenous THAP10 protein complexes and (iii) characterize the potential links between THAP10 and its targets in regulating specific promoters involved in tyrosine kinase signalling pathways.

# Materials and Methods

### Cell lines and patient samples

HL-60, NB4, ME-1, Molt-4, THP-1, MM6, K5652, MV4-11, Jurkat, RAJI, U937, U937-A/E-HA, Kasumi-1, SKNO-1, SKNO-1-siA/E and KG1 cell lines were maintained in RPMI 1640 medium supplemented with 50 μg/ml streptomycin, 50 IU penicillin and 10% FCS. SKNO-1 cells were infected with the pRRLcPPT.hPGK, the lentiviral vector encoding the previously described siAGF1 oligonucleotides (Appendix Table S7) against the *AML1-ETO* mRNA fusion site

(siA/E-RNAs) for silencing of *AML1-ETO* (Fazi *et al*, 2007a; Martinez Soria *et al*, 2009; Li *et al*, 2013b). The U937-A/E-HA clone was obtained by transfection via electroporation of U937 wild-type (WT) cells with an HA-tagged *AML1-ETO* cDNA subcloned into a vector carrying the $Zn^{2+}$-inducible mouse MT-1 promoter, as previously reported (Fazi *et al*, 2007b). Human embryonic kidney (HEK) 293T cells were cultured in DMEM medium supplemented with 50 μg/ml streptomycin, 50 IU penicillin and 10% FCS. Treatments with 5-azacytidine (2.5 μM, Sigma-Aldrich), decitabine (2.5 μM, Xi'an Janssen Pharmaceutical Ltd.) and chidamide (3.0 μM, Shenzhen Chipscreen Biosciences Ltd.) were performed for 40 h.

This study was approved by the Human Subject Ethics Committee in Chinese PLA General Hospital and conducted in accordance with the Declaration of Helsinki. Informed consent was obtained from each subject. Mononuclear cells from bone marrow samples of newly diagnosed, untreated AML patients and healthy donors were isolated using Ficoll-Hypaque (Sigma-Aldrich, St Louis, MO) gradient centrifugation. Patient characteristics are summarized in Appendix Tables S1, S4 and S5.

## RNA and protein extraction and analysis

Total RNA was extracted from cells using the TriPure isolation system (Roche). The relative quantity of *THAP10* mRNA was measured using 200 ng of total RNA by qRT–PCR with qRT–PCR Detection Kit (Ambion, Applied Biosystems, Milan) and the ABI PRISM 7500 Sequence Detection System (Applied Biosystems). Gene expression was determined by the comparative $C_T$ method using *ABL* levels for normalization as recommended in the manufacturer's instructions. Whole cell proteins were extracted using RIPA lysis buffer. Nuclear and cytoplasmic proteins were separated using the NE-PER Nuclear and Cytoplasmic Extraction Kit (Thermo Scientific, Waltham, MA) as per the manufacturer's instructions.

## Immunofluorescence and confocal microscopy

Analysis for nuclear localization of endogenous THAP10 was performed in transfected cells [NIH3T3 cells transfected with pEGFP or p*THAP10*-EGFP; cells were grown on poly-D-lysine-coated cover slips (BD Biosciences, San Jose, CA, USA) 24 h after transfection] or Kasumi-1, HL-60 and t(8;21) AML blasts. Briefly, cells were fixed for 20 min with 4% cold paraformaldehyde and then permeabilized for 5 min with 0.05% Triton X-100. Cells were incubated with 10 μl of Hoechst33342 (100 mg/l, Thermo Fisher Scientific) for 15 min for transfected cells. For Kasumi-1, HL-60 and t(8;21) AML blasts, permeabilized cells were then blocked with PBS-BSA (PBS with 1% bovine serum albumin) and incubated with THAP10 primary antibodies (2 μg/ml, NBP1-86226, Novus Biologicals, USA) overnight at 4°. Cells were then washed three times in PBA-BSA and incubated in the dark for 1 h with Alexa Fluor 488-conjugated secondary antibodies (1/500, #4412, Cell Signalling Technology, USA) diluted in PBA-BSA. After three PBS washes, nuclei were counterstained with DAPI (0.2 μg/ml, #4083, Cell Signalling Technology, USA). Following extensive washing in PBS, samples were air-dried and mounted in Rubber Cement (Union Rubber, Inc., Trenton, NJ, USA). Immunofluorescence images were captured using confocal microscopy (LSM 880, ZEISS, Germany). Green, GFP-*THAP10* or endogenous THAP10; blue, nucleus.

## DNA methylation microarray for 450 k CpG sites

Microarray-based DNA methylation analysis with 450 k arrays was used to profile genome-wide DNA methylation in primary cells, using an Infinium Human Methylation 450 BeadChip with an Infinium HD methylation assay kit and HiScanSQ (Illumina), which assesses the methylation status at more than 485,000 individual CpG sites, encompassing 99% of reference sequence genes and 96% of CpG islands.

Total DNA was extracted using Wizard a genomic DNA purification kit (Promega Corp., WI, USA). DNA quality was determined with a Nano Drop 2000c spectrophotometer (Thermo Fisher Scientific Inc., Wilmington, DE), and 1 μg of DNA was bisulphite converted using a EZ DNA methylation gold kit (Zymo Research Corp., Irvine, CA) according to the manufacturer's protocol. Then, 500 ng of the bisulphate-converted DNA was assayed with an Infinium Human Methylation 450 BeadChip using the Infinium HD methylation assay kit (Illumina), which assesses the methylation status at more than 485,000 individual CpG sites encompassing 99% of reference sequence genes and 96% of CpG islands. Each DNA sample first underwent an overnight isothermal whole-genome amplification step. Amplified DNA was then fragmented, precipitated and resuspended. Samples were hybridized to BeadChips overnight at 48°C in an Illumina hybridization oven. Using an automated protocol on the TecanEvo robot (Tecan Group Ltd., Mannedorf, Switzerland), hybridized arrays were processed through a single-base extension reaction on the probe sequence using DNP- or biotin-labelled nucleotides, with subsequent immunostaining. The BeadChips were then coated, dried and imaged on an Illumina HiScanSQ. Image data were extracted using the Genome Studio version 2010.3 methylation module (Illumina). Beta values were calculated at each locus (β = intensity of methylated allele/intensity of unmethylated allele + intensity of methylated allele + 100), followed by analysis with the R package to normalize the data. The X and Y chromosome methylation data were excluded from the results. Quality control inclusion valuation depended on hybridization detection $P < 0.05$ (Accession number: GSE80508). Differential methylation loci were shown in a volcano plot: red points ($P < 0.05$, FC ≥ 2), and green points ($P < 0.05$, FC ≤ −2) indicate hypermethylation and hypomethylation loci, respectively. Furthermore, a heatmap was generated to demonstrate the indentified genes with abnormal expression/methylation profiles. DAVID online tools were used to implement the enrichment and pathway analysis. Venn diagrams were used to reveal the genes with hypermethylation profiles: These genes are both transcription factor and target of AML1-ETO. The hypermethylation list was derived from the differential analysis of our 450K BeadChips, while transcription factors and targets of AML1-ETO were from animal TFDB database (Zhang *et al*, 2015) and Gardini *et al* (2008), respectively. All statistical analyses were carried out with R 3.2.3.

## Cell transfection

For siRNA-mediated gene knock-down, $1 \times 10^6$ Kasumi-1 cells or SKNO-1 cells were seeded into 6-well plate for overnight incubation before transfection. miRNAs and siRNAs used in the study were synthesized by GenePharma (Shanghai, China). The sequences of siRNA or miRNA are provided in Appendix Table S7. Transfection

with siRNAs and miRNAs was performed using HiPerFect (Qiagen, Hilden, Germany) according to the manufacturer's protocol. In brief, for each well, 24 μl HiPerFect and 12 μl siRNA or 20 μM miRNA were added to 400 μl serum-free RPMI-1640. The mixture was added to cells with 400 μl complete medium and incubated for 6 h, and then, 1.6 ml of complete medium was replenished. The final concentration of miRNA or siRNA was 100 nM. Total RNA and protein were prepared 48 h after transfection and subjected to qRT–PCR or Western blot analysis. Transfection efficiency was determined using FAM-labelled control RNA by a FACScalibur flow cytometer (Becton Dickinson, Franklin Lakes, NJ, USA). This reagent yielded at least 90% transfection efficiency.

For gene overexpression, $2 \times 10^5$ 293 T cells were seeded into a 6-well plate for overnight incubation before transfection. Cells were transfected with a total of 0.5 μg expression plasmids or the corresponding empty vectors using Lipofectamine™ 2000 reagent (Life Technologies) as per the manufacturer's instruction. For co-transfection experiments, the total amount of DNA or siRNA was kept the same between the individual gene and co-transfection groups by adding equal amounts of empty-vector or negative control.

### THAP10-expressing lentivirus preparation and transduction

Lentiviruses expressing EGFP and THAP10 were purchased from HanBio (www.hanbio.net). For lentivirus constructs, the CDS of THAP10 was inserted into hU6-MCS-PGK-EGFP lentiviral vectors (Hanbio, Shanghai, China). The recombinant lentiviruses were produced by co-transfection of 293T cells with the plasmids PSPAX2 and PMD2G with LipoFiter™ (Hanbio, Shanghai, China). Lentivirus-containing supernatants were harvested 48 h after transduction and filtered through 0.22-μm cellulose acetate filters (Millipore, USA). Recombinant lentiviruses were then concentrated by ultracentrifugation (2 h at 50,000 × g). To establish stable cell lines, Kasumi-1 cells were transduced with lentiviral vector with or without THAP10 at an MOI of approximately 100 in the presence of 5 μg/ml polybrene. Approximately 48 h after transduction, the medium was changed, and puromycin was added for selection of stable transduced cells. Puromycin-resistant colonies were selected for 3 weeks and then expanded. Gene modification via lentiviral delivery was confirmed by Western blot analysis.

### Chromatin immunoprecipitation (ChIP)

Three biological replicate ChIP-seq experiments were conducted for the specific detection of AML1-bound genomic regions on THAP10 and miR-383 promoters according to standard procedures with several modifications. Briefly, cross-linked chromatin from approximately $5–10 \times 10^7$ SKNO-1, SKNO-1-siA/E, U937 and U937-A/E cells was prepared and fragmented to an average size of approximately 200 bp by 30–40 cycles of sonication (30 s each) in 15 ml tubes using the Bioruptor UCD-200 sonicator (Diagenode). For immunoprecipitation, AML1 (sc-8563, Santa Cruz Biotechnology), ETO (sc-9737, Santa Cruz Biotechnology), HDAC1 (ab7028, Abcam), DNMT1 (ab13537, Abcam), DNMT3a (ab13888, Abcam), DNMT3b (ab13604, Abcam) and p300 (ab14984, Abcam) antibodies were added to 12 ml of diluted, fragmented chromatin. Nonimmunized rabbit serum served as a control. ChIP using the normal

mouse IgG (Abcam) antibody was performed on naked and soni-cated DNA extracted from the same cell samples and assayed using the EZ-ChIP™ Chromatin Immunoprecipitation kit (Millipore) as per the manufacturer's instructions. Genomic THAP10 and pre-miR-383 upstream regions close to the putative AML1-binding site were amplified. Primer sequences are shown in Appendix Table S7. GAPDH served as a control for nonspecific precipitated sequences.

### Immunoprecipitation and Western blot analysis

Kasumi-1 cells were treated with or without C646 (10 and 20 μM) for 24 h, after which cells were lysed in RIPA buffer containing 1 mM DTT, 1 μg/ml DNase I and a proteinase inhibitor cocktail (Roche) and then incubated on ice for 30 min. Anti-ETO agarose (Santa Cruz Biotechnology) was used for immunoprecipitation assays. For Western blot analysis, protein samples were separated by electrophoresis on denaturing 10% SDS–PAGE gels and blotted to PVDF membranes (Millipore). Immunoblotting was performed using monoclonal antibodies, including anti-AML1 (MABD126, Calbiochem), anti-p-Tyr (sc-51688, Santa Cruz Biotechnology) and anti-Ac-histone H3 (Lys9/14, sc-8655, Cell Signalling Technology). Anti-β-actin (sc-47778, Santa Cruz Biotechnology) was used as loading control. The proteins were visualized by the ECL method (Amersham Biosciences).

### Transactivation assays

DNA fragments were amplified by PCR from human genomic DNA. Primer sequences are shown in Appendix Table S7. All fragments were inserted in pGL3-LUC reporter vectors (Promega). Various mutants were generated using a QuikChange Lightning Site-Directed Mutagenesis kit as per the manufacturer's instructions (Quik-Change, Agilent Technologies). All constructs were verified by DNA sequencing. $2 \times 10^5$ HEK293T cells were plated in 24-well plates and transiently co-transfected by SuperFect (Qiagen) with 10, 50 or 100 ng of pcDNA3 vectors with or without AML1-ETO cDNA, together with 400 ng of the LUC reporter constructs as described above. Co-transfection with a pRL-TK Renilla luciferase reporter vector (Promega) was used as an internal control. Cells were harvested at 48 h post-transfection and analysed using the Dual Luciferase Assay (Promega) as per the manufacturer's instructions.

### Bisulphite sequencing assay

Bisulphite sequencing assays were performed to examine the methylation status of the CpG dinucleotides within the promoter region of the THAP10 gene, using EpiTect Bisulfite kit (Qiagen) as per the manufacturer's instructions. The assay was performed using 1 μg of bisulphite-treated genomic DNA from the indicated cell lines and blasts from untreated patients. After bisulphite conversion performed with EpiTect Bisulfite kit (Qiagen) as previously described (Yu et al, 2005; Li et al, 2013b), the fragments of interest were amplified. Primer sequences are shown in Appendix Table S7. PCR products were gel-purified and cloned into the pGEM®-T vector systems (Promega). Individual bacterial colonies were used for PCR using vector-specific primers, and PCR products were sequenced for the analyses of DNA methylation.

## Luciferase assays for reverse screening the miRNAs regulating *THAP10*

To screen the miRNAs that directly regulate expression of *THAP10*, the 3′-UTR or its mutant was cloned into a modified pGL-3 control vector downstream of the coding sequence of luciferase. The transfection mixtures contained 100 ng of firefly luciferase reporter and 400 ng of plasmids expressing *miR-383* primary transcripts, derived from a constructed miRNA expression library. Transfection with pRL-TK (Promega) was used as an internal control. The firefly luciferase activities were analysed using the Dual Luciferase Assay (Promega) at 48 h after transfection.

## Microarray analysis and functional enrichment of differentially expressed gene (DEGs)

Total RNA was isolated from Kasumi-1 cells after transduction with lentivirus expressing *THAP10* or control using an RNeasy Mini kit (QIAGEN) according to the protocol. RNA quality, concentration and integrity were assessed by an Agilent 2100 Bioanalyzer. Sample labelling, microarray hybridization and washing were performed as per the manufacturer's instructions. Briefly, total RNA was transcribed to double-stranded cDNA, synthesized into cRNA and labelled with cyanine 3-CTP. The labelled cRNAs were then hybridized onto the microarray. After washing, the arrays were scanned by the Agilent Scanner G2505C (Agilent Technologies). Gene expression was analysed with Affymetrix U133 plus2.0 Human Gene Expression Arrays (Affymetrix Technologies). Data collection and analysis were performed according to the protocols by the Gene Biotechnology Corporation (Shanghai, China). DEGs of three independent experiments were used for biological process enrichment with Gorilla, an online gene ontology analyser. In addition, DEGs were used for pathway enrichment using DAVID functional annotation bioinformatics microarray.

## Cell differentiation, apoptosis and colony-forming analysis

Cell differentiation was evaluated by morphological examination of Wright-Giemsa-stained cytospins using light microscopy. Cells were also stained with anti-CD11b (BD Biosciences PharMingen) and then subjected to flow cytometric analysis (a minimum of 50,000 events were collected for each sample) by FACScan (Becton Dickinson) using CellQuest software (Becton Dickinson) for data acquisition and analysis. For cell proliferation assays, cells posttransfection were seeded into 96-well plates (5,000 cells/well), and cell proliferation was then assessed using the CCK-8 kit (Dojindo, Japan) as per the manufacturer's instructions. Briefly, 10 μl of CCK-8 solution was added to the cell culture and incubated for an additional 4 h. Absorbance was determined at 450 nm wavelength. For apoptosis analysis, cells were stained with Annexin V and 7AAD (BD Biosciences) and analysed by a FACSCalibur flow cytometer (Becton Dickinson). Colony formation assays were performed using the methylcellulose H4230 culture system (Stem Cell Technologies) as per the manufacturer's instructions. After incubation for 10 days at 37°C and 5% $CO_2$ in a humidified atmosphere, colonies of > 50 cells with morphological haematopoietic characteristics were counted.

### The paper explained

**Problem**
Aberrant distribution of promoter DNA methylation occurring in specific and distinct patterns has been shown to be a universal feature in t(8;21) AML patients. However, the mechanisms that mediate these aberrant methyl cytosine patterns have not been defined.

**Results**
We conducted a large-scale DNA methylation profiling study and found strong evidence that AML1-ETO contributes to aberrant promoter DNA methylation patterning in t(8;21) AML. We found that the promoter DNA methylation signature of t(8;21) AML blasts differs from those of normal CD34[+] bone marrow cells and non-t(8;21) AMLs. This signature contains 408 differentially methylated genes, many of which are involved in cancer pathways and myeloid leukaemia. Then, database systematic survey and differential methylated regions (DMR) analysis were performed, which demonstrated that a novel hypermethylated zinc finger-containing protein, THAP10, is a target gene and can be epigenetically suppressed by AML1-ETO at the transcriptional level in t(8;21) AML, and suppression of *THAP10* predicts a poor clinical outcome. Our findings also showed that *THAP10* is a *bona fide* target of *miR-383* that can be epigenetically activated by AML1-ETO. In this study, we demonstrated that epigenetic suppression of *THAP10* is the mechanistic link between AML1-ETO fusion proteins and tyrosine kinase cascades. In addition, we showed that THAP10 is a nuclear protein that inhibits myeloid proliferation and promotes differentiation both *in vitro* and *in vivo*.

**Clinical impact**
Our results identified *AML1-ETO/THAP10/miR-383* as a novel epigenetic mini-circuitry in t(8;21) AML, thereby providing a new insight into the mechanism of *AML1-ETO* in driving leukaemogenesis. It also offers an opportunity to develop *THAP10* and *miR-383* as novel biomarkers and therapeutic targets in this AML subtype. Finally, genome-wide DNA methylation profiling is a powerful tool for characterization and clinical stratification of AML and warrants attention in clinical practice.

## Animal studies

BALB/c nude female mice (5 weeks old and weight > 15 g) were utilized. All mice used in this study were bred and maintained in a pathogen-free environment. Splenectomy was performed at least 10 days before inoculation. After the mice recovered from the incisions, general irradiation at a dose of 400 cGy was performed at 24 h before tumour cell inoculation. The mice were then randomly divided into two groups. Xenograft models were established by subcutaneous injection into the right flank with $2 \times 10^7$ Kasumi-1 cells ($n = 4$) or Kasumi-1-*THAP10* cells ($n = 5$). The mice were sacrificed on day 28, and tumour weights were measured. A xenograft model was also established by subcutaneous injection with $2 \times 10^7$ SKNO-1 cells into the right flank of nude mice. Tumour size was measured daily until it reached 50 mm³, after which 5 μg of synthetic *miR-383* or scramble diluted in Lipofectamine solution (Invitrogen) with a total volume of 100 μl was injected directly into tumours on days 1, 3, 7 and 10. Tumours were measured on the day of the first treatment and 4 days after the last treatment when the animals were sacrificed. Necropsies were performed, and tumours were weighed. Tumour volumes were calculated using the equation V (mm³) = A×B²/2, where A is the longest diameter and B is the perpendicular diameter.

## Statistical analysis

SPSS 13.0 software was used to analyse the data. Wilcoxon signed-rank test was performed to determine the differences in *THAP10* and *miR-383* expression in clinical samples. Seventy-six patients were grouped into quartiles according to *THAP10* expression levels and divided into high (*THAP10*-H, $n = 56$) and low (*THAP10*-L, $n = 20$) groups based on the trend observed in clinical outcome after performing a Cox regression analysis of event-free survival (EFS) and overall survival (OS) with *THAP10* quartile grouping as the independent variable. Multivariate survival analysis was carried out using a Cox regression model. Student's *t*-test was used to determine the statistical significance of experimental variables, and *P*-values < 0.05 were considered statistically significant.

Expanded View for this article is available online.

## Acknowledgements

This work was supported by the National Natural Science Foundation of China (Grants 81370635, 81570137, 81270611, 81470010 and 81670162) and the Beijing Natural Science Foundation (Grant 7151009). We thank Professor Qishou Xu for experimental support, and Ms. Li Cheng for technical assistance.

## Author contributions

YL, QN, JS and YC performed experiments and analysed the data; MJ constructed plasmids and detected luciferase; LG, WH and YJ provided AML patient samples and clinical data; SH and AL performed ChIP experiments; ZH, DL and LW performed statistical analysis; CN, YD and MQZ commented on the paper; YL wrote the paper; YL and LY designed the research; and LY supervised the work.

## Conflict of interest

The authors declare that they have no conflict of interest.

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
