## [Review Process File · EMBO Molecular Medicine]

Manuscript EMM-2016-07180

A novel epigenetic AML1-ETO/THAP10/miR-383 mini-circuitry contributes to t(8;21) leukemogenesis

Yonghui Li, Qiaoyang Ning, Jinlong Shi, Yang Chen, Mengmeng Jiang, Li Gao, Wenrong Huang, Yu Jing, Sai Huang, Anqi Liu, Zhirui Hu, Daihong Liu, Lili Wang, Clara Nervi, Yun Dai, Michael Q Zhang, Li Yu

Corresponding author: Li Yu & Yonghui Li, Chinese PLA General Hospital

Review timeline:

Submission date:	08 October 2016
Editorial Decision:	18 November 2016
Revision received:	15 March 2017
Editorial Decision:	04 April 2017
Revision received:	11 April 2017
Accepted:	21 April 2017

Editor: Roberto Buccione

Transaction Report:

1st Editorial Decision

18 November 2016

Thank you for the submission of your manuscript to EMBO Molecular Medicine. We have now heard back from the three Reviewers whom we asked to evaluate your manuscript.

We are very sorry that it has taken longer than usual to get back to you on your manuscript. We experienced significant difficulties in recruiting appropriate reviewers and then in obtaining the evaluations in a timely fashion.

As you will see, the reviewers all appear to express general appreciation for the interest and value of the findings reported, albeit with different degrees of enthusiasm. However, they also clearly point to many unclear and/or poorly supported conclusions and the need to provide substantial reworking of the manuscript at various levels. I will not dwell into much detail as the evaluations are quite thorough, but I would like to highlight the main points.

For instance reviewer 1, among other important concerns, would like you to perform a significant amount of validation in primary cells from patients and to extend your analysis to the recently published AML methylomes, which in part would address the concern expressed by reviewer 2 on the databases used. Reviewer 2, in addition to the above, also finds that the conclusion that THAP10 repression is crucial for leukemogenesis is not currently supported and criticizes, as does reviewer 3, the use of NIH3T3 cells. Reviewer 2, similarly to reviewer 1 also highlights the need for additional mechanistic insight. Reviewer 3, in addition to other points, also wonders how you came about to investigate miRNA in the regulation of THAP1p expression, as does reviewer 2.

In conclusion, while publication of the paper cannot be considered at this stage, given the potential interest of your findings and after further reviewer cross-commenting and internal discussion, we have decided to give you the opportunity to address the criticisms. Please consider that the concerns raised are of great importance for us as they impinge on the overall quality and robustness of experimental support for the main conclusions.

We are thus prepared to consider a substantially revised submission, with the understanding that the reviewers' concerns must be addressed with additional experimental data where appropriate and that acceptance of the manuscript will entail a second round of review. I can understand that provision of validation data with human primary cells might be difficult, but I agree with the reviewer that such data would significantly improve the clinical impact of your work. I therefore encourage you to develop your study in this sense as far as realistically possible for your next, revised version

Please note that it is EMBO Molecular Medicine policy to allow a single round of revision only and that, therefore, acceptance or rejection of the manuscript will depend on the completeness of your responses included in the next, final version of the manuscript.

EMBO Molecular Medicine now requires a complete author checklist (<http://embomolmed.embopress.org/authorguide#editorial3>) to be submitted with all revised manuscripts. Provision of the author checklist is mandatory at revision stage; The checklist is designed to enhance and standardize reporting of key information in research papers and to support reanalysis and repetition of experiments by the community. The list covers key information for figure panels and captions and focuses on statistics, the reporting of reagents, animal models and human subject-derived data, as well as guidance to optimise data accessibility.

As you know, EMBO Molecular Medicine has a "scooping protection" policy, whereby similar findings that are published by others during review or revision are not a criterion for rejection. However, I do ask you to get in touch with us after three months if you have not completed your revision, to update us on the status. Please also contact us as soon as possible if similar work is published elsewhere.

Please note that we now mandate that all corresponding authors list an ORCID digital identifier. You may do so through our web platform upon submission and the procedure takes <90 seconds to complete. I would also encourage your co-authors to supply an ORCID identifier, which will be linked to their name for unambiguous name identification including for future publications.

I look forward to seeing a revised form of your manuscript as soon as possible.

***** Reviewer's comments *****

Referee #1 (Remarks):

In the present manuscript, the authors suggest that DNA methylation the promoters in t(8;21) AML blasts differs from other AMLs. They suggest a novel hyper-methylated zinc finger containing protein-THAP10 is a target gene being epigenetic silenced by AML1-ETO. THAP10 is target of miR-383, which can be epigenetic activated by AML1-ETO recruiting p300. The authors suggest that the epigenetic silencing of THAP10 is a mechanistic link between AML1-ETO fusion proteins and tyrosine kinase cascades, being THAP10 able to inhibit myeloid proliferation and promoting differentiation both in vitro and in vivo.

Finally, the authors indicate the existence of an epigenetic mini-circuitry of AML1-ETO/THAP10/miR-383 in t(8;21) AML, in which silencing of THAP10 predicts a poor clinical outcome.

Major points

a. Methylome analyses. The authors identify THAP10 as silenced by DNA methylation in AML1-ETO AMLs. In Fig. 1 they show the analyses. Since data from 200 methylomes have been published in AMLs (Ley et al. NEJM 2013) from the TCGA, it is crucial to compare the data from this study

to the data included in the Ley et al. study (data are freely available upon request; some can be downloaded even without). The same applies for gene expression related to THAP10 in AML. Data should be included reporting THAP10 expression in the 200 Ley et al. AMLs.

b. The authors indicate that THAP10 gene silencing is associated with unfavorable outcome of t(8;21) AML patients. They show that in 76 AML1-ETO patients, THAP10 gene expression is inverse correlated with survival. This observation is supported by different experimental approach that identify an AML1-ETO containing complex at the THAP10 promoter. All the experiments should include data of primary AML AML1-ETO+ and AML1-ETO- blasts (with particular reference to Fig. 3D).

c. Epidrug treatment. The authors show that treatment with 5-Aza increased THAP10 expression by ~4.0-fold in AML1/ETO+ cells displaying strong hypermethylation of THAP10 (Figure 3E), co-treatment with 5-Aza and the HDAC inhibitor TSA further increases its expression (Figure S4C). The experiment with the epidrugs has been performed in cells lines. It would be strengthening the data to add ex vivo primary blasts treated in culture with hypomethylating agents and HDACis. In addition, the authors might include HDACi which have been approved or are in clinical trials such as Vorinostat, Faridak, Istodax etc. If available, data on blasts from patients which might have been enrolled in clinical trials with HDACi or hypomethylating agents might be very interesting (if available).

d. miR383, AML1-ETO, THAP10. The authors suggest that AML1-ETO forms a complex with p300 at the AML1-binding sites in the pre-miR-383 regulatory region contributing to transcriptional activation of miR-383, which in turn negatively regulates THAP10 expression in t(8;21) AML. The authors show that the p300 inhibitor C646 is able to decrease acetylation in this context. Is p300i (C646) able to decrease miR383 expression? If so, is p300i (C646) able to increase THAP10 expression? How these data (p300i and HDACi have opposite actions) might be integrated with the fact that HDACi increase THAP10 expression?

e. The authors support a notion that the nuclear protein THAP10 inhibits proliferation but promotes differentiation of t(8;21) AML cells. In addition,

f. The experiments in support mainly based on gain and lost of THAP10 in AML cells lines are descriptive. The authors should try to include mechanistic hypothesis on how THAP10 might act. How THAP10 has the capability to increase differentiation? Which mechanism can be hypothesized and proven? Is the expression of THAP10 higher in differentiated myeloid cells?

g. In vivo data. The authors might include experiments by using anti-miR-383 or scramble oligonucleotides in AML-ETO- AML cells in support of the specificity of its activity for AML1-ETO+ cells.

Minor points

- The sentence in the abstract 'we found that the promoter DNA methylation signature of t(8;21) AML blasts differs from those of other AMLs' should be clarified.
- In figure 2C, THAP10 mRNA levels were measured in blasts of AML1/ETO+ patients who achieved remission after chemotherapy; these levels were higher than the same patients at relapse. How this experiment has been done exactly? If the patients were in complete remission, how blasts were found and taken? Were (or not) patients in CR?
- The authors indicate that miR-383 expression was decreased in blasts of AML1/ETO+ patients who achieved complete remission following induction chemotherapy, compared to the same patients at relapse (Figure 4E). Again, how this experiment has been performed? How and how many blasts where found in CR patients? Or were these blasts taken before?
- The authors argue that THAP10 is a nuclear protein. Data on the endogenous protein localization in AML1-ETO blasts should be included.

Referee #2 (Remarks):

In this study, Li and colleagues describe the effects of downregulating THAP10, a nuclear factor that inhibits myeloid proliferation and promotes differentiation in t(8;21) AML. This repression is initiated by AML1-ETO, the fusion oncogene produced in this specific type of AML, by two separate mechanisms: one based on epigenetic modifications, and the another, on miRNAs. Overall, this study is of interest, as it reveals a new factor (THAP10) implicated in AML. However, there are still a few experiments missing would increase the impact of this manuscript if included.

Major points

The first part of the results is very interesting. The aberrant methylation of AML1-ETO+ as compared to AML1-ETO- in normal bone marrow (NBM) cells seems consistent, and several candidates that could mediate this are identified. However, the authors use "publicly-available databases" for these results, so that it is not clear how the nine factors depicted in Fig. 1D compare with their own data. The sources of the databases should be also better highlighted.

The second part of the results starts with the identification of THAP10 as one of the most hypermethylated targets in AML-ETO+. It would be important to state whether the other top 9 factors are also found in the databases depicted in Fig. 1D.

The results shown for THAP10 are well-done: it is characterized in both cell lines and patient-derived cells, and its correlation with AML1-ETO is shown as well as its direct regulation are nicely documented. Nonetheless, we are left wondering if the remaining 8 factors also have AML1-ETO sites in their promoters.

In Fig. 3A, the Katsumi-1 cell line is used to demonstrate that THAP10 is reduced at the RNA level (as shown by analyzing AML1-ETO expressing cell), but that there is not much difference at the protein level between HL-60 and Katsumi-1. The authors should comment on this.

It seems that the THAP10 is targeted by miR-383. Although the data presented are interesting, it is initially unclear why the authors decided to investigate the contribution of miRNAs in this context; later in the manuscript, the authors state that AML1-ETO induces the expression of miR-383. For the reader, the jump in elucidating the mechanism is confusing without a transition sentence or rearrangement of results.

The data indicate that miR-383 anti-correlates with THAP10, both in cell lines and in patient samples. However, the relation between miR-383 and THAP10 is not strongly established, and Fig 4B should be repeated and quantified. Additional experiments to establish a direct proof of miR-383 effects on THAP10 mRNA should be included.

How can the authors conclude that THAP10 repression is crucial for leukomogenesis? Its localization was investigated in NIH-3T3 cells rather than in leukemic cells. The authors perform expression analysis upon overexpression of THAP10 in Katsumi-1 cells, and they correlate the phenotype (cell growth) with deregulation of several signaling pathways and cell differentiation. Are the same set of genes also upregulated when the knockdown of THAP10 is performed in HL60 cells, where its expression level is high? Since HL60 cells do not have the AML1-ETO fusion protein but do have high levels of THAP10, can this cellular model really be compared with Katsumi-1 cells? In other words, is there a general correlation between THAP10 levels and aggressiveness of leukemic cell lines, irrespective of the presence of AML1-ETO?

The discussion, as well as the last part of the introduction, is repetitive. It would be useful for the authors to discuss why they think there are two complementary mechanisms to repress THAP10. It would be also important to put in context their findings with other factors described as main players of AML1-ETO repression, such as the RARbeta receptor.

Minor

In general, English language revision is necessary.

Please check and correct any confusions between knock-out and knockdown (for example, manuscript line 192).

Lines 196/197: there seems to be a mistake in the description of the condition with 5-Aza in Fig. 3E.

Referee #3 (Comments on Novelty/Model System):

Technical quality is high because the authors present a large amount of experiments to support their claims. The novelty is high because this is the first time a direct link between AML1-ETO, THAP10 and miR-383 has been described. Clearly this link describes a double edged targeting of THAP10 by AML1-ETO, resulting in its down regulation, which inhibits the differentiation of the leukemic cells. Some modeling of THAP10 regulation on tumorigenesis is performed in immunodeficient mice.

Referee #3 (Remarks):

The authors describe a novel epigenetic AML1-ETO/THAP10/miR-383 regulatory circuit in AML1-ETO myeloid leukemia. They provide convincing evidence for the existence of this circuit using patient expression data and biochemical verification.

Comments:

Figure 1A: Which are the rows for the normal CD34+ cells and which are the two t(8;21) sub-clusters? As presented it is impossible to distinguish which is which. The authors should include an explanation of the color marking in the figure legend.
What does high versus low THAP10 mRNA expression mean in Figs. 1G, H and I, i.e. what is 'high' and what is 'low'?

Fig. 3A: AML1-ETO expression is higher in Kasumi than in SKNO-1 but THAP10 expression is also higher than in SKNO-1. How do the authors explain this contradiction?

How does expression of exogenous AML1-ETO in U937-A/E compare to the endogenous expression of AML1-ETO in SKNO-1 (Fig. 3D)? Is it similar or much higher?

This reviewer would like to see an anti-p-TYR blot of Kasumi cells expressing vector or THAP10 in addition to Fig 6C, to show the overall down regulation tyrosine phosphorylated proteins in Kasumi-THAP10 cells.

What was the reason to look for miRNAs that would down regulate THAP10 expression?

Fig 5B AML1-ETO and p300 may bind to closely spaced binding sites in the miR-383 promoter but that doesn't prove that they form a complex. Therefore, the presented data do not support the conclusion in lines 261-264. Did the authors try to co-IP AML1-ETO with p300?

Line 67: It is stated that THAP10 is present in PML nuclear bodies but the authors do not show a THAP10/PLM colocalization experiment. Without that the authors cannot claim that THAP10 is present in PML NBs, given that there different, non-overlapping puncta in the nucleus. Moreover, the punctate staining might be a result of overexpression of THAP10 in the NIH3T3 cells. Therefore the authors should show THAP10 localization in HL60, which they show expresses a decent amount of endogenous protein (Fig. S7A).

Minor comments:

Line 322: Fig. S6C should be Fig. S7C.

Line 192: The authors use AML-ETO knockdown to diminish THAP10 CpG methylation not knockout as is written in the text.

Line 202: Based on the presented data AML1-ETO does not silence THAP10 but rather suppresses its expression.

Line 274: Were the Kasumi cells transfected or transduced with the Lenti-T10 viral vector? The correct terminology to introduce lentivirus in cells is transduction. Please change the wording throughout the text where appropriate.

Lines 327 and 328: Fig. S6D should be S7D.

Line 343: Epigenetic silencing should be epigenetic down regulation because THAP 10 is still expressed but at a lower level.

"Hoechst" is misspelled in Fig. 6A and the accompanying text.

Referee #1 (Remarks):

1. Methylome analyses. The authors identify THAP10 as silenced by DNA methylation in AML1-ETO AMLs. In Fig. 1 they show the analyses. Since data from 200 methylomes have been published in AMLs (Ley et al. NEJM 2013) from the TCGA, it is crucial to compare the data from this study to the data included in the Ley et al. study (data are freely available upon request; some can be downloaded even without). The same applies for gene expression related to THAP10 in AML. Data should be included reporting THAP10 expression in the 200 Ley et al. AMLs.

Response: We thank the reviewer for this critical recommendation. In response, we have compared the results of the present study (**Figure 1D**) to the new results (new **Figure EV 1**) obtained from an additional analysis that incorporated the data from the TCGA methylomes (7 cases for AML-ETO+ M2 and 37 cases for AML-ETO- M2; Ley et al. NEJM,2013) with the data from the A/E Chip and Animal TFBD, the same publically-available databases employed in our study. As shown in **Figure EV 1**, 4 genes (highlighted by red font) were in common when compared the data from these two analyses, which includes THAP10. As requested, we have also performed another analysis for THAP10 mRNA level in the 200 AML samples (Ley et al. NEJM 2013) using the data from the TCGA by the Boxplot analysis. It was found that THAP10 mRNA was the lowest in t(8;21)+ AML, compared to all other subtypes (new **Figure EV 2**), consistent with our results (**Figure 2B**). Thus, the results obtained from these new analyses of the data from the TCGA database further support our claim that THAP10 is silenced by DNA methylation in AML1-ETO AML.

2. The authors indicate that THAP10 gene silencing is associated with unfavorable outcome of t(8;21) AML patients. They show that in 76 AML1-ETO patients, THAP10 gene expression is inverse correlated with survival. This observation is supported by different experimental approach that identify an AML1-ETO containing complex at the THAP10 promoter. All the experiments should include data of primary AML AML1-ETO+ and AML1-ETO- blasts (with particular reference to Fig. 3D).

Response: This comment is valid. We agree with the reviewer that it is important to validate our results, which were obtained from human AML cell lines, in primary AML blasts. In response, we have conducted additional experiments using the CHIP assay in primary AML blasts. As shown in new **Figure EV 3C**, the results from 2 primary AML samples were analogous to those observed previously in human leukemia cell lines (**Figure 3D**). While the luciferase reporter assay requires transfection with two or more plasmids, highly efficient transfection of primary cells remains however a technological challenge. Thus, we felt that this assay is unfortunately not suitable for primary cells such as AML blasts.

3. Epidrug treatment. The authors show that treatment with 5-Aza increased THAP10 expression by ~4.0-fold in AML1/ETO+ cells displaying strong hypermethylation of THAP10 (Figure 3E), co-treatment with 5-Aza and the HDAC inhibitor TSA further increases its expression (Figure S4C). The experiment with the epidrugs has been performed in cells lines. It would be strengthening the data to add *ex vivo* primary blasts treated in coculture with hypomethylating agents and HDACi. In addition, the authors might include HDACi which have been approved or are in clinical trials such as Vorinostat, Faridak, Istodax etc. If available, data on blasts from patients which might have been enrolled in clinical trials with HDACi or hypomethylating agents might be very interesting (if available).

Response: Again, this point is valid. In response, we have conducted additional experiments involving clinically-relevant hypomethylating agents (Decitabine) and/or HDACi (Chidamide). As shown in new **Figure EV 3E** and **3F**, treatment with Decitabine +/- Chidamide also clearly increased *THAP10* expression *ex vivo* in primary blasts as well as *in vivo* in patients enrolled in a ongoing clinical trial (NCT02886559) .

4. miR383, AML1-ETO, THAP10. The authors suggest that AML1-ETO forms a complex with p300 at the AML1-binding sites in the pre-miR-383 regulatory region contributing to transcriptional activation of miR-383, which in turn negatively regulates THAP10 expression in t(8;21) AML. The authors show that the p300 inhibitor C646 is able to decrease acetylation in this context. Is p300i (C646) able to decrease miR383 expression? If so, is p300i (C646) able to

increase THAP10 expression? How these data (p300i and HDACi have opposite actions) might be integrated with the fact that HDACi increase THAP10 expression?

Response: We apologize for this confusion. In fact, one of the interesting findings in the present study is that while HDAC1 was found at the AML1-binding sites of THAP10 gene (**Figure 3D** and **new Figure EV 3C**), it is however absent in the miR-383 promoter region (**Figure 5C**). These results argue that HDACs (e.g., HDAC1) might have direct effect on expression of THAP10, but not miR-383, which promoted our attempt to find out how miR-383 is regulated by AML1-ETO. We found a marked increase in binding of the histone acetyltransferase (HAT) p300 to the miR-383 promoter region, in close association with AML1-ETO expression (**Figure 5C**). These results suggest that AML-ETO might act to regulate THAP10 expression via two separate processes: a) directly via recruitment of the co-repressor HDAC1 to the promoter of THAP10 gene; b) indirectly by increasing p300 bound to the promoter of miR-383 and thereby activating expression of miR-383, which in turn suppressed THAP10 expression. Thus, these dual mechanisms might explain why both HATi (e.g., C646; **Figure 5F**) and HDACi (e.g., TSA or Chidamide; **new Figure EV 3D-F**) increased THAP10 expression, despite their opposite effects on histone acetylation. In response, we have now clarified these points on pages 12 (2nd paragraph, for HDACi) and 16 (2nd paragraph, for p300i).

5. The authors support a notion that the nuclear protein THAP10 inhibits proliferation but promotes differentiation of t(8;21) AML cells.

In addition, the experiments in support mainly based on gain and lost of THAP10 in AML cells lines are descriptive. The authors should try to include mechanistic hypothesis on how THAP10 might act. How THAP10 has the capability to increase differentiation? Which mechanism can be hypothesized and proven? Is the expression of THAP10 higher in differentiated myeloid cells?

Response: We thank the reviewer for the valid recommendation to include a mechanistic hypothesis for how THAP10 acts to promote cell differentiation (e.g., morphologic changes and increased CD11b+ cells after overexpressing THAP10, **Figure 6H** and **6I**), a phenomenon noted in our study. However, it is a big challenge and time-consuming work to find out and then prove the mechanism(s) by which THAP10 promotes cell differentiation, due to lack of understanding of the THAP10's function in general and virtually none in AML. Therefore, while this observation is interesting, we felt that addressing in depth the mechanism underlying this role of THAP10 would most appropriately be conducted in a successor study due to time constraints for resubmission as well as the length of the present manuscript. As requested, we have now included a hypothesis that "loss of function" of the transcriptional repressor THAP10 may result in the activation of a tyrosine kinase pathway, which therefore promotes cell proliferation but blocks cell differentiation of myeloid progenitors in Discussion (page 24-25). We have also added a statement that the mechanism for THAP10-mediated cell differentiation remains to be defined (page 25).

6. In vivo data. The authors might include experiments by using anti-miR-383 or scramble oligonucleotides in AML-ETO- AML cells in support of the specificity of its activity for AML1-ETO+ cells.

Response: Again, we appreciate the reviewer's kind suggestion. As requested, we have conducted additional in vivo experiments by using anti-miR-383 or scramble oligonucleotides in mice bearing tumors derived from AML-ETO⁻ AML (HL-60) cells. As shown in new **Figure EV 5E and F**, there was no significant difference in tumor size between administration of anti-miR-383 and scramble oligonucleotides ($P = 0.700$). These results indicate that unlike in mice bearing AML-ETO⁺ tumor (SKNO-1 cells; **Figure 7E-H**), targeting miR-383 failed to reduce tumor growth in mice carrying AML-ETO⁻ tumor (HL-60), suggesting the selectivity for the role of miR-383 in AML-ETO⁺ AML.

7. The sentence in the abstract 'we found that the promoter DNA methylation signature of t(8;21) AML blasts differs from those of other AMLs' should be clarified.

Response: We apologize for this vagueness. In response, we have modified this sentence by clarifying other AMLs as t(8;21)⁻ AMLs in Abstract (page 3).

8. In figure 2C, THAP10 mRNA levels were measured in blasts of AML1/ETO+ patients who achieved remission after chemotherapy; these levels were higher than the same patients at relapse. How this experiment has been done exactly? If the patients were in complete remission, how blasts were found and taken? Were (or not) patients in CR?

Response: This is a valid comment. The reviewer is correct that there is few blasts in peripheral blood and/or bone marrow of patients who achieve complete remission. We apologize for this error.

In response, we have now corrected "blasts" as "mononuclear cells" on page 10, as well as clarified "samples" as "mononuclear cells isolated from bone marrow samples" in the legend of **Figure 2C**.

- The authors indicate that miR-383 expression was decreased in blasts of AML1/ETO+ patients who achieved complete remission following induction chemotherapy, compared to the same patients at relapse (Figure 4E). Again, how this experiment has been performed? How and how many blasts were found in CR patients? Or were these blasts taken before?

Response: Again, we apologize for this error. In response, we have now corrected "blasts" as "mononuclear cells" on page 14, as well as clarified "samples" as "mononuclear cells isolated from bone marrow samples" in the legend of **Figure 4E**.

- The authors argue that THAP10 is a nuclear protein. Data on the endogenous protein localization in AML1-ETO blasts should be included

Response: We apologize for this vagueness. Nuclear localization of THAP10 was determined under fluorescence microscopy after transfection of NIH3T3 cells with a GFP-tagged THAP10 vector (**Figure 6A**). In response, we have performed additional experiments to confirm this result, including a) by Western blot analysis of endogenous THAP10 protein expression in nuclear (NE) versus cytoplasmic (CE) fractions (new **Figure EV 4E**); and b) by immunofluorescent staining for endogenous THAP10 protein in AML1-ETO⁺ (e.g., Kasumi cells) versus AML1-ETO⁻ (e.g., HL-60 cells), as well as in primary t(8;21)AML blast (new **Figure EV 4F**). The results from these experiments support the notion that THAP10 is indeed a nuclear protein.

Referee #2 (Remarks):

- The first part of the results is very interesting. The aberrant methylation of AML1-ETO+ as compared to AML1-ETO- in normal bone marrow (NBM) cells seems consistent, and several candidates that could mediate this are identified. However, the authors use "publicly-available databases" for these results, so that it is not clear how the nine factors depicted in Fig. 1D compare with their own data. The sources of the databases should be also better highlighted.

Response: We appreciate the reviewer's encouraging comment that our results are interesting. We also apologize for this vagueness about how this analysis was done. In fact, the Venn diagrams presents the results from an analysis by incorporating our own data (i.e., 408 differentially methylated genes in t(8;21) AML, **Figure 1B**) with the data from the publicly-available databases, which identified 9 candidate gene that might be targeted by AML1-ETO (**Figure 1D**). In response, we have modified the text to clarify this point on page 8 as well as the legend of **Figure 1D**. As requested, we have also highlighted the sources of the databases employed in this analysis by citing new references, including the databases for genes (Gardini et al, 2008) and for transcription factors (Animal TFBD, Zhang et al, 2015).

- The second part of the results starts with the identification of THAP10 as one of the most hypermethylated targets in AML-ETO+. It would be important to state whether the other top 9 factors are also found in the databases depicted in Fig. 1D.

Response: This comment is valid. As requested, we have included a statement that except THAP10, other top 9 of 10 genes listed in **Appendix Table S3** were not found in the set of genes depicted in **Figure 1D**, on page 9 (2nd paragraph).

- The results shown for THAP10 are well-done: it is characterized in both cell lines and patient-derived cells, and its correlation with AML1-ETO is shown as well as its direct regulation are nicely documented. Nonetheless, we are left wondering if the remaining 8 factors also have AML1-ETO sites in their promoters.

Response: Again, we appreciate the reviewer's positive comment that the experiments for characterizing THAP10 are well-done and its correlation with and regulation by AML1-ETO nicely documented. As 9 genes depicted in **Figure 1D** were identified by an analysis that surveys the genes fulfilled two criteria as follows: a) those targeted by AML1-ETO; and b) those that function as transcription factors. Thus, all these 9 candidate genes are supposed to have AML1-ETO-binding sites in their promoter. However, we are not able to draw such a conclusion without further confirmation like we did for THAP10. In response, we have now included this possibility in Discussion (page 23, 1st paragraph).

4. In Fig. 3A, the Katsumi-1 cell line is used to demonstrate that THAP10 is reduced at the RNA level (as shown by analyzing AML1-ETO expressing cell), but that there is not much difference at the protein level between HL-60 and Katsumi-1. The authors should comment on this.

Response: The reviewer is correct that despite their significant difference at mRNA levels, the difference in protein levels of THAP10 was unclear between HL-60 and Katsumi-1 cells. In response, we have repeated this Western blot analysis, which again showed a moderate, although visible, difference between these two lines (**Figure 3A**). This inconsistency between mRNA and protein levels suggests two possibilities that a) AML1-ETO might not be able to completely silencing the expression of THAP10 in AML1-ETO+ cells like Kasumi-1, implying that other factors may also be involved, likely to a lesser extent, in transcriptional regulation of THAP10 expression; b) THAP10 protein level might be also determined by other mechanism(s) e.g., via protein turnover. In response, we have included a comment on this phenomenon on page 11 (2nd paragraph).

5. It seems that the THAP10 is targeted by miR-383. Although the data presented are interesting, it is initially unclear why the authors decided to investigate the contribution of miRNAs in this context; later in the manuscript, the authors state that AML1-ETO induces the expression of miR-383. For the reader, the jump in elucidating the mechanism is confusing without a transition sentence or rearrangement of results.

Response: We appreciate the reviewer's comment that the data for miR-383 is interesting. We also apologize for this confusion. In fact, in addition to the epigenetic silencing of THAP10 gene expression by AML1-ETO, we further tested whether other mechanism(s) would also be involved in transcriptional regulation of this gene in AML1-ETO AML cells. Indeed, we found that *miR-383* also plays a functional role to link AML1-ETO with suppression of *THAP10* expression. Therefore, two separate, although related, mechanisms, by which AML1-ETO acts to suppress THAP10 expression (i.e., direct methylation-mediated silencing of the THAP10 promoter and indirect suppression of THAP10 expression via up-regulation of inhibitory miR-383), were identified in the present study. In response, we have included a transition sentence to link these two parts on page 14.

6. The data indicate that miR-383 anti-correlates with THAP10, both in cell lines and in patient samples. However, the relation between miR-383 and THAP10 is not strongly established, and Fig 4B should be repeated and quantified. Additional experiments to establish a direct proof of miR-383 effects on THAP10 mRNA should be included.

Response: This comment is valid. In response, we have repeated the Western blot experiment shown in **Figure 4B**, which again showed a moderate reduction of THAP10 after treated with synthetic *miR-383*. As requested, we have also quantified the THAP10 blots as shown in **Figure 4B**. We have also conducted additional experiments to validate the functional role of miR-383 in regulation of THAP10 expression as follows: a) luciferase reporter assay, which revealed a ~50 reduction of luciferase activity of wild-type THAP10 3'-UTR, but not miR-383-binding site mutated THAP10 3'-UTR, by miR-383 (new **Figure EV 4A**); and b) qPCR, which indicated a significant reduction of THAP10 mRNA expression in HL-60 and NB4 cells treated with synthetic *miR-383*, compared to untreated and scramble microRNA controls (**Figure EV 4B**). Together, we felt that these results provide direct evidence supporting a notion that miR-383 negatively regulates THAP10 expression, particularly at mRNA level.

7. How can the authors conclude that THAP10 repression is crucial for leukemogenesis? Its localization was investigated in NIH-3T3 cells rather than in leukemic cells. The authors perform expression analysis upon overexpression of THAP10 in Katsumi-1 cells, and they correlate the phenotype (cell growth) with deregulation of several signaling pathways and cell differentiation. Are the same set of genes also upregulated when the knockdown of THAP10 is performed in HL60 cells, where its expression level is high? Since HL60 cells do not have the AML1-ETO fusion protein but do have high levels of THAP10, can this cellular model really be compared with Katsumi-1 cells? In other words, is there a general correlation between THAP10 levels and aggressiveness of leukemic cell lines, irrespective of the presence of AML1-ETO?

Response: This is a valid comment. We agree with the reviewer that it is risky to draw a conclusion that THAP10 repression is crucial for leukemogenesis, only based on the present evidence that overexpression of THAP10 suppressed tumor growth in a model using Katsumi cells with low basal level of THAP10. Due to lack of understanding of THAP10's function in general and particularly in leukemia, we have not found any evidence supporting a correlation between THAP10 levels and aggressiveness of leukemia, although AML1-ETO has been well documented as a marker for poor

prognosis of patients with AML. In response, we have modified the text throughout to avoid such a conclusion.

8. The discussion, as well as the last part of the introduction, is repetitive. It would be useful for the authors to discuss why they think there are two complementary mechanisms to repress THAP10. It would be also important to put in context their findings with other factors described as main players of AML1-ETO repression, such as the RARbeta receptor.

Response: In response, we have re-written the last part of Introduction by removing the repetitive content. We have also discussed why we thought there might be more than one mechanisms involved in repression of THAP10 expression by AML1-ETO on page 24 (2nd paragraph). As requested, we have acknowledged that other factors such as RAR β 2 (a tumor suppressor gene) has been reported to be epigenetically silenced in t(8;21) AML (page 23, 2nd paragraph), with a new reference (Fazi et al, 2007b).

9. In general, English language revision is necessary.

Response: In response, we have polished language with assistance of special agencies.

10. Please check and correct any confusions between knock-out and knockdown (for example, manuscript line 192).

Response: We apologize for these confusions. As the shRNA approaches were used to down-regulate expression of target genes, we have corrected this term as “knock-down” throughout the text, to avoid a confusion with cells obtained from gene knock-out mice.

11. Lines 196/197: there seems to be a mistake in the description of the condition with 5-Aza in Fig. 3E.

Response: We apologize for this error. We have corrected "**Figure 3E**" as (new **Figure EV 3D**).

Referee #3 (Comments on Novelty/Model System):

We appreciate the reviewer's comments that technical quality and novelty of the present study are high, as well as this is the first time a direct link between AML1-ETO, THAP10, and miR-383 has been described.

(Remarks)

1. Figure 1A: Which are the rows for the normal CD34+ cells and which are the two t(8;21) sub-clusters? As presented it is impossible to distinguish which is which. The authors should include an explanation of the color marking in the figure legend.

Response: We apologize for this indistinct presentation of the heatmap in **Figure 1A**. In response, we have included additional squares to cluster each category of the samples. As requested, we have also included the explanation of the color markers that label the three categories of the samples in the legend of Figure 1A.

2. What does high versus low THAP10 mRNA expression mean in Figs. 1G, H and I, i.e. what is 'high' and what is 'low'?

Response: We respectively think that the reviewer refers to Figure 2G, H and I. We apologize for missing this definition. In response, we have included the cut-off value (e.g., 2×10^1 of THAP10 expression) by which high versus low THAP10 mRNA expression was defined, in legend for **Figure 2G-2I**. In addition, we have also added a dash line to indicate the cut-off value in these figures.

3. Fig. 3A: AML1-ETO expression is higher in Kasumi than in SKNO-1 but THAP10 expression is also higher than in SKNO-1. How do the authors explain this contradiction?

Response: This is a valid point. In fact, another reviewer also raised a similar question about this inconsistency, which we felt, is at least in part due to poor quality of these blots (particularly for the lanes of SKNO-1 WT and Mock). In response, we have repeated this Western blot experiment. As shown in new **Figure 1A**, the levels of AML1-ETO and THAP10 were actually much closer between Kasumi and SKNO-1 cell lines. We apologize for this error.

4. How does expression of exogenous AML1-ETO in U937-A/E compare to the endogenous expression of AML1-ETO in SKNO-1 (Fig. 3D)? Is it similar or much higher?

Response: We apologize for this confusion. We also agree with the reviewer that it is difficult to compare the results from SKNU-01 cells, which have high endogenous AML1-ETO, to those stemmed from ectopic overexpression of exogenous AML1-ETO in U937 cells, which are lack of endogenous AML1-ETO. Actually, in this experiment, we attempted to compare SKNO-1 cells with its AML1-ETO shRNA counterparts (labeled as SKNO-1-siA/E), and U937 cells with its AML1-ETO overexpressing counterparts (labeled as U937-A/E), respectively. In response, we have clarified these comparisons in the text (page 12, 1st paragraph).

5. This reviewer would like to see an anti-p-TYR blot of Kasumi cells expressing vector or THAP10 in addition to Fig 6C, to show the overall down regulation tyrosine phosphorylated proteins in Kasumi-THAP10 cells.

Response: This comment is valid. We agree with the reviewer that it is important to validate the findings on mRNA expression of target genes by Western blot analysis to examine their protein levels as well as post-translational modifications (e.g., tyrosine phosphorylation of target proteins). As required, we have detected the total tyrosine phosphorylation by western blot analyses and assessed a diminution of about 50% in p-Tyr levels in *THAP10*-expressing Kasumi-1 cells as compared with mock-transduced cells (**Appendix Figure S5**).

6. What was the reason to look for miRNAs that would down regulate THAP10 expression?

Response: The regulation levels of genes include transcription, post transcription, translation and post translation et. al. In transcription level, we shown that AML1-ETO directly bound to and hypermethylated the promoter region of *THAP10*, resulting in epigenetic silencing of this gene. In translation level, we shown that *miR-383* suppress the expression of *THAP10*, also resulting in epigenetic silencing of this gene. Interesting, AML1-ETO cooperated with p300 and acetylated the promoter of *miR-383*. There was a mini-circuitry exist among *AML1-ETO*, *miR-383* and *THAP10*. What's more, miRNAs plays an important role in the lineage differentiation of hematopoietic cells by regulating expression of oncogenes or tumor suppressors. Deregulation of miRNA expression has been shown to be involved in multistep carcinogenesis and has rapidly emerged as a novel therapeutic target. We have added this transition sentence in context.

7. Fig 5B AML1-ETO and p300 may bind to closely spaced binding sites in the miR-383 promoter but that doesn't prove that they form a complex. Therefore, the presented data do not support the conclusion in lines 261-264. Did the authors try to co-IP AML1-ETO with p300?

Response: This is a valid comment. We agree with the reviewer that without direct evidence for binding of AML1-ETO with p300 (e.g., a co-IP analysis), it is risky to draw a conclusion that they form a complex in the region of the miR-383 promoter. In response, we have now modified the text to avoid such a claim (page 17, 2nd paragraph).

8. Line 67: It is stated that THAP10 is present in PML nuclear bodies but the authors do not show a THAP10/PLM colocalization experiment. Without that the authors cannot claim that THAP10 is present in PML NBs, given that there different, non-overlapping puncta in the nucleus. Moreover, the punctate staining might be a result of overexpression of THAP10 in the NIH3T3 cells. Therefore the authors should show THAP10 localization in HL60, which they show expresses a decent amount of endogenous protein (Fig. S7A).

Response: We apologize for this vagueness. The phenomenon that THAP1 is present in PML nuclear bodies was observed previously by Cayrol et al (Cayrol et al, 2007). We here cite this earlier work to raise a possibility that THAP10, as another member of the THAP family that includes THAP1, may also be a nuclear protein that is required for its transcription-regulatory activity. In response, we have now modified the text to clarify this point (page 5). As requested, we have performed addition experiment of immunofluorescence staining for THAP10 in HL-60 cells. As shown in new **Figure EV 4F**, the endogenous THAP10 protein was primarily localized in the nuclei, consistent with our observation in NIH3T3 cells (**Figure 6A**).

9. Line 322: Fig. S6C should be Fig. S7C.

Response: We apologize for this error. This citation has now been corrected have corrected (now as new **Figure EV 5C**) on page 21 (line 355).

10. Line 192: The authors use AML-ETO knockdown to diminish THAP10 CpG methylation not knockout as is written in the text.

Response: As requested, we have corrected “knock-out” to “knock-down” (page 12, line 194).

11. Line 202: Based on the presented data AML1-ETO does not silence THAP10 but rather suppresses its expression.

Response: We have corrected "silencing" to "suppressing" (page 12, line 193) as requested.

12. Line 274: Were the Kasumi cells transfected or transduced with the Lenti-T10 viral vector? The correct terminology to introduce lentivirus in cells is transduction. Please change the wording throughout the text where appropriate.

Response: We have corrected “transfected” to “transduced” where lentivirus was used to introduce gene into cells, throughout the manuscript.

13. Lines 327 and 328: Fig. S6D should be S7D.

Response: Again, we apologize for this error. The citation has now been corrected have corrected (now as new **Figure EV 5D**) on page 21 (line 360).

14. Line 343: Epigenetic silencing should be epigenetic down regulation because THAP 10 is still expressed but at a lower level.

Response: As requested, we have corrected “epigenetic silencing” to “epigenetic suppression” for THAP10 expression throughout the text.

15. "Hoechst" is misspelled in Fig. 6A and the accompanying text.

Response: We apologize for this typo, which has now been corrected in Figure 6A as well as in the text (page 29, line 496).

2nd Editorial Decision

04 April 2017

Thank you for the submission of your revised manuscript to EMBO Molecular Medicine. We have now received the enclosed reports from the referees that were asked to re-assess it. As you will see the reviewers are now globally supportive and I am pleased to inform you that we will be able to accept your manuscript pending the following final amendments:

1) Please improve the scale bars in Fig EV4F. They are currently barely visible.

2) Please remove the "For More Information" section and insert the accession number information in the appropriate position in the Materials and Methods section. I also note that in the "Data Accession" section of your checklist you mention that such information is provided on page 42, which I did not find to be the case.

3) Reviewer 3 notes that the manuscript would benefit from language and text editing. I agree and would strongly recommend that you have the manuscript edited by a native English speaker. If this is not possible or a preferred option, you might consider one of the available language editing services e.g. <http://wileyeditingservices.com>.

Please submit your revised manuscript within two weeks. I look forward to seeing a revised form of your manuscript as soon as possible.

***** Reviewer's comments *****

Referee #1 (Remarks):

The present version of the manuscript addresses the concerns highlighted previously and clarifies some of the major points.

Referee #2 (Remarks):

The authors have addressed all my previous concerns.

Referee #3 (Comments on Novelty/Model System):

Technical quality is high because the authors present a large amount of experiments to support their claims. The novelty is high because this is the first time a direct link between AML1-ETO, THAP10 and miR-383 has been described. Clearly this link describes a double edged targeting of THAP10 by AML1-ETO, resulting in its down regulation, which inhibits the differentiation of the leukemic cells. Some modeling of THAP10 regulation on tumorigenesis is performed in immunodeficient mice.

Referee #3 (Remarks):

No further remarks other than that the English is still subpar.

2nd Revision - authors' response

11 April 2017

Thank you very much for the re-assessment of our revised manuscript "A novel epigenetic AML1-ETO/THAP10/miR-383 mini-circuit contributes to t(8;21) leukaemogenesis" by Li et al., EMM-2016-07180. We appreciate your kind invitation to submit a revised manuscript. We also appreciate reviewers for their constructive comments and globally supports. In our revised manuscript, we have carefully addressed virtually all of the comments made by the editor and reviewers. A point-by-point responses to the comments follows below.

1) Please improve the scale bars in Fig EV4F. They are currently barely visible.

Response: We apologize for this vagueness. In response, we have improved the scale bars in Fig EV 4F.

2) Please remove the "For More Information" section and insert the accession number information in the appropriate position in the Materials and Methods section. I also note that in the "Data Accession" section of your checklist you mention that such information is provided on page 42, which I did not find to be the case.

Response: As requested, we have removed the "For More Information" section and inserted the accession number in the "Materials and Methods" section (Page 32). We also changed such information in our checklist.

3) Reviewer 3 notes that the manuscript would benefit from language and text editing. I agree and would strongly recommend that you have the manuscript edited by a native English speaker. If this is not possible or a preferred option, you might consider one of the available language editing services e.g. <http://wileyeditingservices.com>.

Response: Thanks for your suggestion. This manuscript has been edited for proper English language, grammar, punctuation, spelling, and overall style by one or more of the highly qualified native English speaking editors at Wiley Editing Services.

We appreciate the opportunity to re-revise our manuscript, and hope that the modifications we have made satisfactorily address the reviewer's comments, and make it acceptable for publication in EMBO Molecular Medicine. If there are any remaining questions about our manuscript, please let us know and we'll be glad to answer them.

Corresponding Author Name: Li Yu

Manuscript Number: EMM-2016-07180